# An autocatalytic multicomponent DNAzyme nanomachine for tumor-specific photothermal therapy sensitization in pancreatic cancer

Jiaqi Yan[1,2,3,4,7], Xiaodong Ma[3,4,7], Danna Liang [5,7], Meixin Ran[1,2,3,4,7], Dongdong Zheng [6], Xiaodong Chen[1,2], Shichong Zhou[6], Weijian Sun[5] ✉, Xian Shen[1,2] ✉ & Hongbo Zhang [1,2,3,4] ✉

Multicomponent deoxyribozymes (MNAzymes) have great potential in gene therapy, but their ability to recognize disease tissue and further achieve synergistic gene regulation has rarely been studied. Herein, Arginylglycylas-partic acid (RGD)-modified Distearyl acylphosphatidyl ethanolamine (DSPE)-polyethylene glycol (PEG) (DSPE-PEG-RGD) micelle is prepared with a DSPE hydrophobic core to load the photothermal therapy (PTT) dye IR780 and the calcium efflux pump inhibitor curcumin. Then, the MNAzyme is distributed into the hydrophilic PEG layer and sealed with calcium phosphate through biomineralization. Moreover, RGD is attached to the outer tail of PEG for tumor targeting. The constructed nanomachine can release MNAzyme and the cofactor $Ca^{2+}$ under acidic conditions and self-assemble into an active mode to cleave heat shock protein (HSP) mRNA by consuming the oncogene miRNA-21. Silencing miRNA-21 enhances the expression of the tumor suppressor gene PTEN, leading to PTT sensitization. Meanwhile, curcumin maintains high intracellular $Ca^{2+}$ to further suppress HSP-chaperone ATP by disrupting mito-chondrial $Ca^{2+}$ homeostasis. Therefore, pancreatic cancer is triple-sensitized to IR780-mediated PTT. The in vitro and in vivo results show that the MNAzyme-based nanomachine can strongly regulate HSP and PTEN expression and lead to significant pancreatic tumor inhibition under laser irradiation.

Mild photothermal therapy can avoid inflammation and tumor metastasis caused by excessive hyperthermia during cancer therapy[1]. However, the curative effect of mild PTT is dramatically compromised by heat shock proteins (HSPs), which are produced by the cellular thermal defense mechanism[2,3]. The combination of HSP small molecule inhibitors and photothermal agents has been considered an effective PTT strategy[4,5]. Nevertheless, small molecule inhibitors normally suffer from obvious hysteretic effects, since they can only inhibit already generated HSPs, rather than prevent the production of HSPs

before treatment[6]. Furthermore, small molecule inhibitors will inevitably inhibit HSPs in peritumoral tissues; therefore, unavoidable thermal diffusion during PTT will intensify the side effects of tumor treatment[7]. More importantly, targeting not only HSPs but also PTT-associated targets such as ATP[8,9] and miRNA-21[10] could also lead to a synergistic effect. Therefore, achieving multilevel PTT sensitization in specific cancer cells is of great significance.

Oligonucleotide therapeutic agents can regulate target genes intracellularly by means of supplementing downregulated genes[11] or

by sterically blocking or cleaving overexpressed genes[12] and possess irreplaceable advantages compared with traditional drugs[13]. A deoxyribozyme (DNAzyme) is a single-stranded DNA-based oligonucleotide therapeutic agent[14] that contains a catalytic core flanked by two substrate arms that can be endowed with immense biocatalytic functions by employing various metal cofactors[15,16]. Leveraging excellent programmability and intrinsic biocompatibility, DNAzymes can be constructed as specific-mRNA hydrolysis machines with multiple turnover rates[17]. Thus far, they have been utilized in many advanced applications, including disease biomarker detection[18], intelligent release system construction[19,20], specific mRNA applications[21], or even double-stranded DNA regulation with higher efficiency than the CRISPR–Cas9 system[22]. However, although researchers can accurately design DNAzymes with target gene-silencing functions, the "off-target" effects are still serious since the DNAzymes cannot regulate target genes in specific tissues[23]. For example, under photothermal therapy (PTT), it is very difficult to use DNAzymes to silence HSP70 mRNA only in pancreatic tumor sites while retaining HSP70 function in paracancerous tissues, therefore distinguishing the two sites with different photothermal responsive sensitivities.

Multicomponent deoxyribozymes (MNAzymes) are further engineered inventions from the two most widely used DNAzymes (17E and 10-23)[24], which contain two partzyme strands created by splitting the DNAzyme at the catalytic core (Fig. 1a, b)[25]. Each partzyme comprises half of each substrate's binding arm, target binding arm, and catalytic core[26]. This engineered segmentation process endows MNAzymes with the ability to recognize cancer biomarkers[27,28]. Although the category of target biomarker sequence (miRNA-155[27], miRNA-21[28], miRNA-10b[29], antibiotic resistance genes[30], SARS-CoV-2 virus sequence[31], etc.) has been broadly explored for more than 10 years of MNAzyme development, the MNAzyme substrate strand has been mainly a stereotype for fluorescence/quencher-based signal detection (Fig. 1c) rather than biomarker-responsive target gene silencing (common stereotype sequences of

the cleavage site for 17E-based MNAzyme: 3′-TrArGG-5′[24,25,27,28,32] and 10-23-based MNAzyme: 3′-CrUrGC-5′[25,29,33]). Encouragingly, a recent study has demonstrated the potential of using miRNA for the self-assembly of MNAzymes to achieve tumor-targeted delivery[34]. However, the investigation into the silencing of miRNA to further enhance the therapeutic effect has been neglected. The lack of such research represents a significant gap in the current understanding of this MNAzyme therapeutic concept, which therefore warrants further exploration in future studies.

Therefore, designing an MNAzyme system for mild PTT that considers the targeting effect of the cancer biomarker miRNA-21 and further regulates both miRNA-21[35] and HSP70 mRNA (substrate sequence) could effectively overcome the off-target problem and achieve a significant synergistic PTT effect. Silencing miRNA-21 can upregulate PTEN expression, which in turn sensitizes the tumor to PTT[10], while inhibiting HSP70 can directly reduce the tumor's heat resistance. Notably, hybridization between miRNA and the partzyme target binding arm will silence the miRNA via a steric blocking function rather than RNase H-dependent degradation process[36,37]. This silencing mechanism ensures the stable structure of partzyme/miRNA and guarantees the multiple-turnover substrate cleavage activity of MNAzyme.

Meanwhile, poor oligonucleotide delivery efficiency and low content of metal cofactors in tumor microenvironments also challenge the activity of MNAzyme in vivo for cancer therapy[38]. As a result of the rapid and thriving development of nanomedicine, metal ion cofactors can be utilized for nanocarrier construction, with excellent particle encapsulation efficiency achieved through electrostatic interactions between the DNA phosphate backbone and mineral cofactors[39,40]. In our case, to further enhance PTT sensitization, $Ca^{2+}$ was chosen as a cofactor to activate the MNAzyme system. Calcium ions can also mediate mitochondrial calcium dysregulation, which disrupts the mitochondrial structure and results in the inhibition of the HSP-chaperone molecule ATP[41].

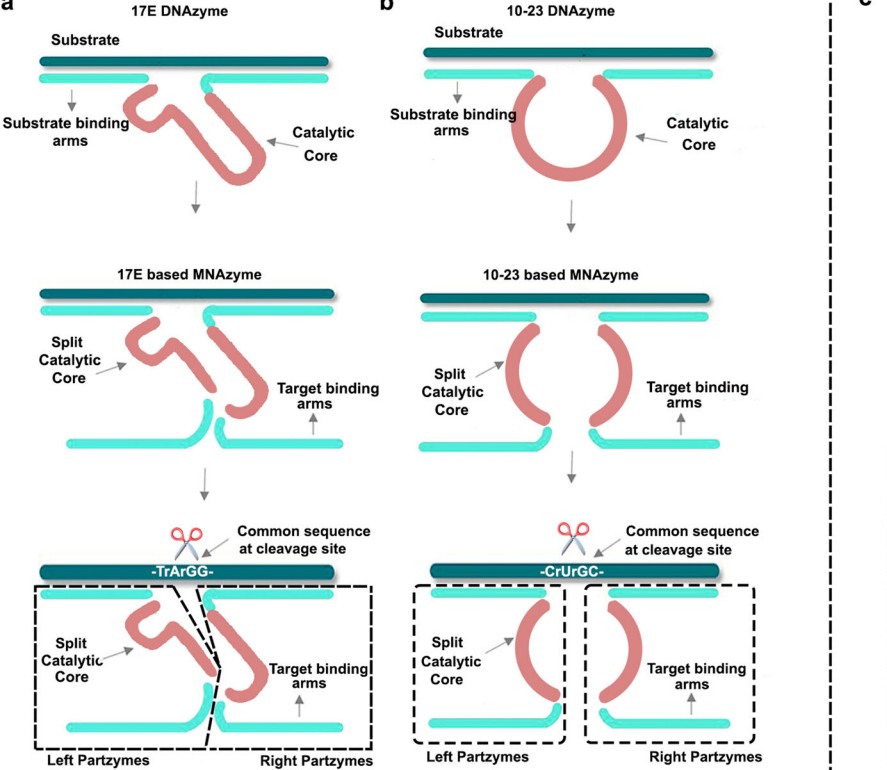

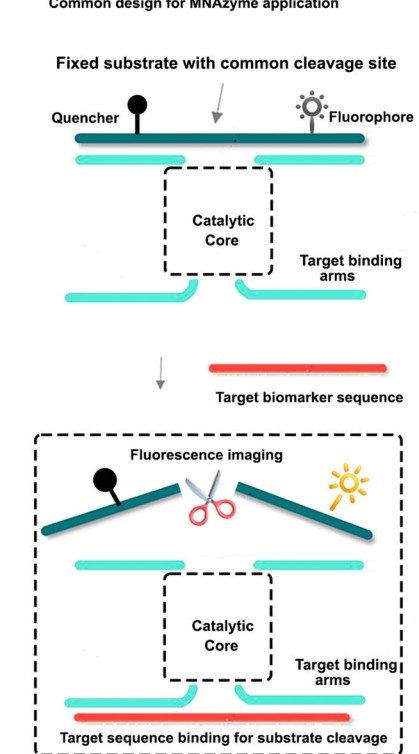

Fig. 1 | **Structures of MNAzymes. a** 17E-based MNAzyme. **b** 10-23-based MNAzyme. **c** Common design for MNAzyme-based applications.

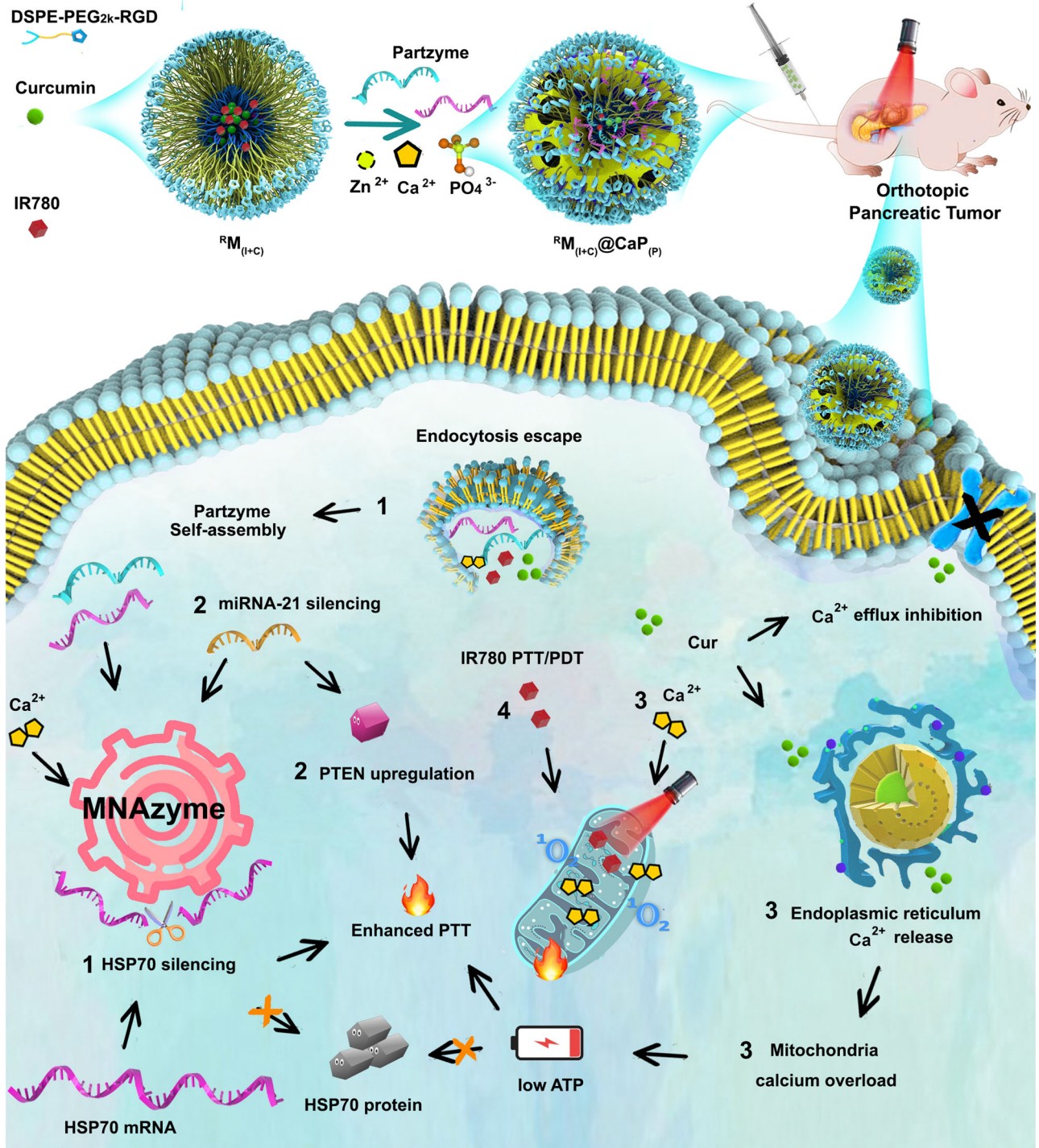

**Fig. 2 | Illustration of four-step reactions after nanosystem fabrication and delivery.** (1): The MNAzyme will self-assemble on-site by consuming miRNA-21 and $Ca^{2+}$ ions and silencing HSP70 mRNA before photothermal therapy (PTT) treatment. (2) Silencing miR-21 causes Phosphatase and TENsin homolog deleted on chromosome 10 (PTEN) upregulation for PTT sensitization. (3) $Ca^{2+}$ will be absorbed by mitochondria, and curcumin will promote the further release of $Ca^{2+}$ from the ER and inhibit the $Ca^{2+}$ efflux pump. Mitochondrial calcification can reduce HSP-chaperones adenosine triphosphate (ATP) production. (4): IR780 will target mitochondria to achieve precise mild PTT treatment and produce ROS-mediated photodynamic therapy.

In this work, we construct a miR-21-targeted MNAzyme-powered HSP70 mRNA cleavage machinery to restrain HSP70 production before PTT treatment. We fabricate a DSPE-PEG$_{2k}$ micelles modified with the tumor-targeting peptide RGD for the coloading of the mitochondrion-targeted PTT dye IR780 and the $Ca^{2+}$ efflux pump inhibition agent curcumin (abbreviated as $^RM_{(I+C)}$). The MNAzyme is mineralized on the PEG layer of the micelle system using calcium phosphate to form the final nanodevice, $^RM_{(I+C)}@CaP_{(P)}$ (Fig. 2). After cell internalization, CaP produce tremendous $Ca^{2+}$ for lysosomal escape. As a result, some enzymes consume miRNA-21 and self-assemble into $Ca^{2+}$-assisted MNAzyme-powered HSP70 mRNA silencing machines in tumor tissues while retaining the HSP70 protection effect in healthy tissues. The silencing of miRNA-21 causes PTEN expression to be upregulated, which also sensitize the tumor tissues to

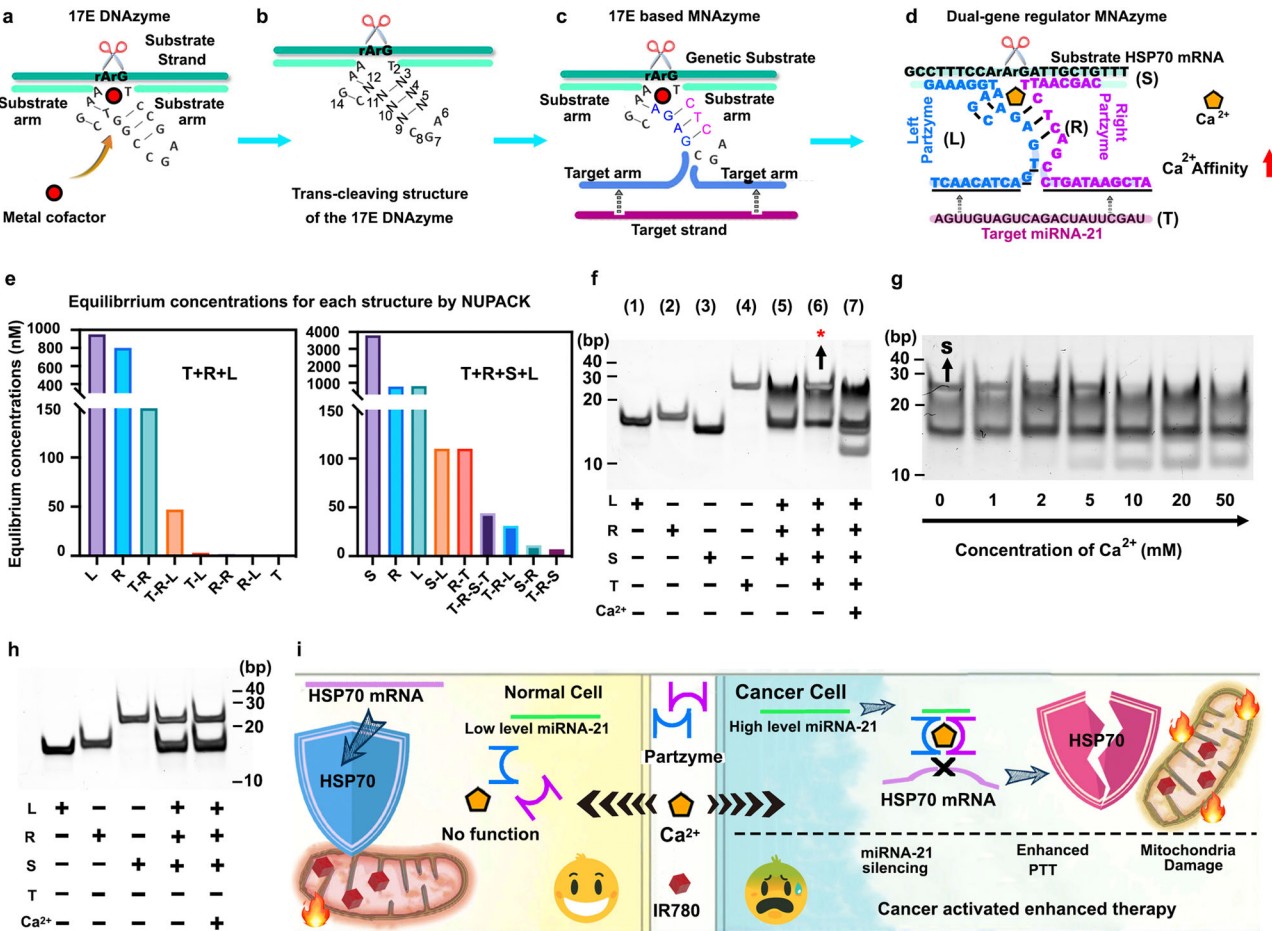

**Fig. 3 | Architecture of the Ca²⁺-responsive MNAzyme system for dual-gene adjustment. a** Secondary structure of the traditional 17E DNAzyme. **b** Variant structures of the 17E DNAzyme. N′ represents the alternative site and can be an A, T, G, or C deoxynucleotide. **c** Structure of the Ca²⁺-responsive 17E-based MNAzyme system. **d** The final MNAzyme system with high Ca²⁺ affinity for miRNA-21 triggered HSP70 mRNA cleavage. **e** The NUPACK web application simulated the equilibrium concentration of each binding chance. **f** Twenty percent polyacrylamide gel electrophoresis experiments for different formulations. **g** Verification of the Ca²⁺ concentration-dependent MNAzyme system. **h** Cancer biomarker response test. **i** Construction logic of the cancer biomarker-responsive MNAzyme system. The experiments in (**f**–**h**) were repeated three times independently with similar results. PTT photothermal therapy, T target strand, R right partzyme, L left partzyme, S substrate mRNA.

increase PTT efficiency. Further sensitization to IR780-mediated PTT is accomplished through Ca²⁺-induced multichannel mitochondria disruption due to curcumin-mediated calcium efflux pump inhibition and Ca²⁺ release from the endoplasmic reticulum (ER) in cancer cells. The addition of zinc prevents the phase transformation of amorphous CaP to hydroxyapatite, enhancing the stability of the nanoformulations during storage. Finally, we demonstrated the ability of the MNAzyme system to selectively regulate tumor tissue and adjacent tissue in an orthotopic pancreatic cancer model.

## Results

To build a miR-21 responsive DNAzyme system for HSP70 mRNA depletion, two common DNAzyme models (17E and 10-23) were chosen. These two DNAzymes are substrate-universal and can be tailored to recognize specific mRNAs by designing the substate arm sequence[42]. Interestingly, the 17E DNAzyme was recently found to be more active than the 10-23 DNAzyme when the same metal cofactor at the same concentration was implemented[43]. The catalytic ring structure of the 17E DNAzyme has also been fundamentally studied (Fig. 3a, b), and the substitution of different sequence fragments in the catalytic ring can affect the catalytic activity and responsiveness of the DNAzyme to different metal ions. For instance, 3-base-pair replacements at the stem position (N³-N⁵, N⁹-N¹¹), as well

as replacement of thymine (T) at the N¹² position adenine (A), can significantly enhance the Ca²⁺ affinity of the 17E DNAzyme (Fig. 3c)[32,44]. Therefore, in this study, 17E DNAzyme was applied to design an efficient Ca²⁺-responsive MNAzyme dual-gene regulation machine.

We first designed the Ca²⁺-specific MNAzyme system by setting the target sequence as one complementary to miRNA-21 (abbreviated as T), while the substrate arm was engineered to match HSP70 mRNA (abbreviated as S). The MNAzyme was split from the catalytic core and divided into the right partzyme (abbreviated as R) and left partzyme (abbreviated as L) (Fig. 3d). The NUPACK web application[45] was utilized to understand the possible reaction scenarios (Fig. 3e). From the final equilibrium system of the T + R + L system (each strand concentration was set as T = 0.2 μM, R = 1 μM and L = 1 μM), T (miRNA-21) showed a strong affinity with R (right partzyme), which formed a 150 nM complex; T and L (left partzyme) showed less combination efficiency (only 3 nM). Surprisingly, T, R and L formed a 47 nM MNAzyme system (T-R-L). These simulation results unexpectedly indicated that the self-building of MNAzyme consists of two processes: T and R achieve the first conjugation, and then L combines with the T-R complex to realize the second step of MNAzyme construction. This step-by-step assembly can theoretically enhance the utilization efficiency of miRNA-21 for multiple-turnover cleavage of HSP70 mRNA in the case of low

partzyme content (Supplementary Fig. 1). Similarly, the T + R + S + L reaction simulation results (L = 1 μM, R = 1 μM, T = 0.2 μM and S = 4 μM) showed that the T-R-L system bound to the substrate S sequence and formed a 44 nM (T-R-S-L) complex. Of note, the T-R-S-L complex exhibited a lower free energy of secondary structure (−38.96 kcal/mol) compared with that of the T-R-L system (−27.84 kcal/mol), which revealed that the T-R-S-L structure was more stable (Supplementary Fig. 1).

Then, the feasibility of MNAzyme-based dual-gene modulation was evaluated in reality by 20% polyacrylamide gel electrophoresis (PAGE) tests (Fig. 3f). The individual L, R, S and T strands all showed clear bands (Lanes 1, 2, 3 and 4), while an obvious new band appeared upon incubation of L, R and T for 30 min at 37°C (L = 1 μM; R = 1 μM; T = 0.2 μM, Lane 5). These results indicated that partzymes possessed strong binding affinity with miRNA-21, which facilitated efficient MNAzyme construction. For a deeper understanding of the multiple turnover catalytic cleavage effect of the MNAzyme system on the HSP70 substrate strand (S), we incubated excess S chain (4 μM) with the abovementioned system, as revealed in Lane 6. A large amount of unreacted S strand appeared at the top of the band (marked with a red asterisk in Fig. 3f, Lane 6). Importantly, the S strand was not cleaved due to the lack of catalysis by the metal ions in Lane 6. Subsequently, when 10 mM $Ca^{2+}$ was added to the system in Lane 7, the substrate chain S was cleaved and disappeared, and a new band was formed from the PAGE test result, as shown in Fig. 3f, Lane 7. These results demonstrated that our designed partzymes could utilize 0.2 μM miRNA to achieve $Ca^{2+}$-mediated 4 μM HSP70 multiple-turnover cleavage and consume all miRNA-21 to achieve dual gene regulation.

To further verify the metal ion dependence of the partzyme system, a substrate catalytic hydrolysis test was performed under different $Ca^{2+}$ concentrations for 4 h (Fig. 3g). L (1 μM) and R (1 μM) strands together with 0.2 μM T strand were applied. We found that as the $Ca^{2+}$ concentration increased from 0 to 50 mM, the content of substrate S (4 μM) gradually decreased, and the product amount consistently increased. Furthermore, the hydrolysis rate of substrate strand HSP70 mRNA reached saturation at ~20 mM $Ca^{2+}$. These results reflected that a high catalyst rate of the MNAzyme system could be guaranteed when sufficient calcium ions were present in the mRNA regulation system.

Ultimately, to demonstrate the cancer responsiveness of the designed partzymes, i.e., their safety in healthy tissues, we carried out another PAGE assessment to investigate the hydrolysis performance of the MNAzyme system for HSP70 mRNA in the absence of miRNA-21 (with 50 mM $Ca^{2+}$) (Fig. 3h). We observed that although in the presence of sufficient $Ca^{2+}$, partzymes could not self-assemble into MNAzymes to mediate the catalytic hydrolysis of HSP70 mRNA. Therefore, hypothetically, when partzymes, $Ca^{2+}$ and a photothermal therapeutic agent (IR780) are simultaneously delivered to the tumor microenvironment (TME), the partzymes can distinctively self-assemble into an MNAzyme-driven HSP70 mRNA-regulating machine in specific miRNA-21-overexpressing pancreatic cancer cells. These processes avoid hysteresis effects by preventing the generation of heat shock proteins at the mRNA level before photothermal treatment. Moreover, during the assembly process of MNAzymes, most of the miRNA-21 within the TME will be consumed, thus realizing dual-gene regulation for both the pathogenic gene miRNA-21 and the therapeutic stimulated gene HSP70 (Fig. 3i).

The miRNA-21-dependent self-construction process of MNAzyme and the specific gene cleaving function under $Ca^{2+}$ triggering have been verified, yet the intracellular activation of the MNAzyme system suffers from poor cellular internalization and low cytoplasmic $Ca^{2+}$ content (~0.1 μM)[46]. Therefore, a smart and robust partzyme delivery strategy using $Ca^{2+}$ as the multicargo carrier component to achieve target site $Ca^{2+}$ enhancement has been explored. Nanostructured amorphous calcium phosphate (CaP) has been broadly investigated as an excellent nuclear acid delivery system owing to its high affinity for the gene phosphate backbone (Fig. 4a)[47], pH-responsive degradability and sustained release performance[48]. However, amorphous calcium phosphate (ACP) composed of Posner's clusters $(Ca_9(PO_4)_6)$ is an unstable precursor that can dissolve and renucleate as hydroxyapatite (HA) within a few minutes in aqueous solution[49,50], causing inefficient cargo delivery. To verify this phenomenon, transmission electron microscopy (TEM) was utilized to systematically observe the crystal transformation of ACP, as shown in Fig. 4b. We found that the ACP was a complete and smooth sphere with irregular channels on its surface within 10 min of preparation. However, the storage of APC in Milli-Q water resulted in the dissolution of ACP (>1 h), nucleation (>3 h) and full crystallization to HA after 48 h. Moreover, we also found that this crystal transformation process could take place in the cell culture medium after 24 h of incubation and that the large size and irregular needle-like shape structure made it difficult to deliver nucleotide drugs and even calcium ions into cells (Fig. 4c).

The zinc additive was found to stabilize the ACP phase for more than 20 days when the Zn/Ca ratio was 10%[51]. In the reaction system in which $Zn^{2+}$ and $Ca^{2+}$ existed simultaneously, the chemical reactions followed the equations below[52]:

$$3Zn^{2+} + 2H_2PO_4^{-} \leftrightarrow Zn_3(PO_4)_2 + 4H^{+} \quad (1)$$

$$3Ca^{2+} + 2H_2PO_4^{-} \leftrightarrow Ca_3(PO_4)_2 + 4H^{+} \quad (2)$$

$$Ca^{2+} + 2Zn^{2+} + 2H_2PO_4^{-} \rightarrow CaZn_2(PO_4)_2 \downarrow + 4H^{+} \quad (3)$$

In this way, highly stable ACP NPs were fabricated, as depicted in Fig. 4d, which contributed to the mechanism by which smaller zinc ions can easily replace $Ca^{2+}$ in ACP and inhibit crystallization by distorting atomic order and reducing ACP solubility[51]. To assess the content of $Ca^{2+}$ and DNAzymes, we used two different assay kits to measure the concentrations of $Ca^{2+}$ and $Zn^{2+}$ ions within the nanosystem, and we found the concentration of zinc ions to be 196 μg and the concentration of calcium ions to be 1280 μg within 3.5 mg Zn-doped CaP NPs.

Nevertheless, $Zn^{2+}$-substituted APC NPs did not possess cancer cell-enhanced endocytosis, and surface pores were reduced by the tight cooperation between zinc and phosphate ions, resulting in the inability to load multiple therapeutics. To resolve the abovementioned difficulties, RGD-modified DSPE-PEG amphiphilic micelles ($^RM$) were used as the synthesis template of $Zn^{2+}$-doped CaP since the negatively charged oxygen atom of the $CH_2$-O-$CH_2$ group in PEG can attract $Ca^{2+}$ or $Zn^{2+}$ through electrostatic attraction (Fig. 4e)[53]. Meanwhile, the micelles can be loaded with multiple hydrophobic drugs, and RGD can realize cancer cell targeting[54].

We first synthesized IR780/Cur-loaded micelles, and a spherical structure of ~50 nm could be seen under TEM (Fig. 4f, $^RM_{(I+C)}$). During preparation, we found that 50 mg of DSPE-PEG-RGD could carry up to 7.98 mg of IR780 and 2.06 mg of Cur. (Tables S1 and S2). To achieve the optimal synergistic treatment ratio of both drugs after being loaded by the nanosystem, the loading weight ratio of IR780:Cur (mg) was adjusted to 0:2, 0.5:2, 1:2, 2:2, 4:2, and 8:2 to calculate the coefficient of drug interaction (CDI)[55,56]. The calculation was based on the WST-1 absorbance results, which compared the viability of cells after treatment with different drug ratios of nanoparticles. The equation is as follows:

$$CDI = AB/(A \times B) \quad (4)$$

where AB is the absorbance of the combination groups to the control group, while A or B is the absorbance of the single-agent group to the control group (CDI > 1, =1, <1 indicates antagonism, additivity and synergism). In accordance with the WST-1 results, a 4:2 ratio (4 mg

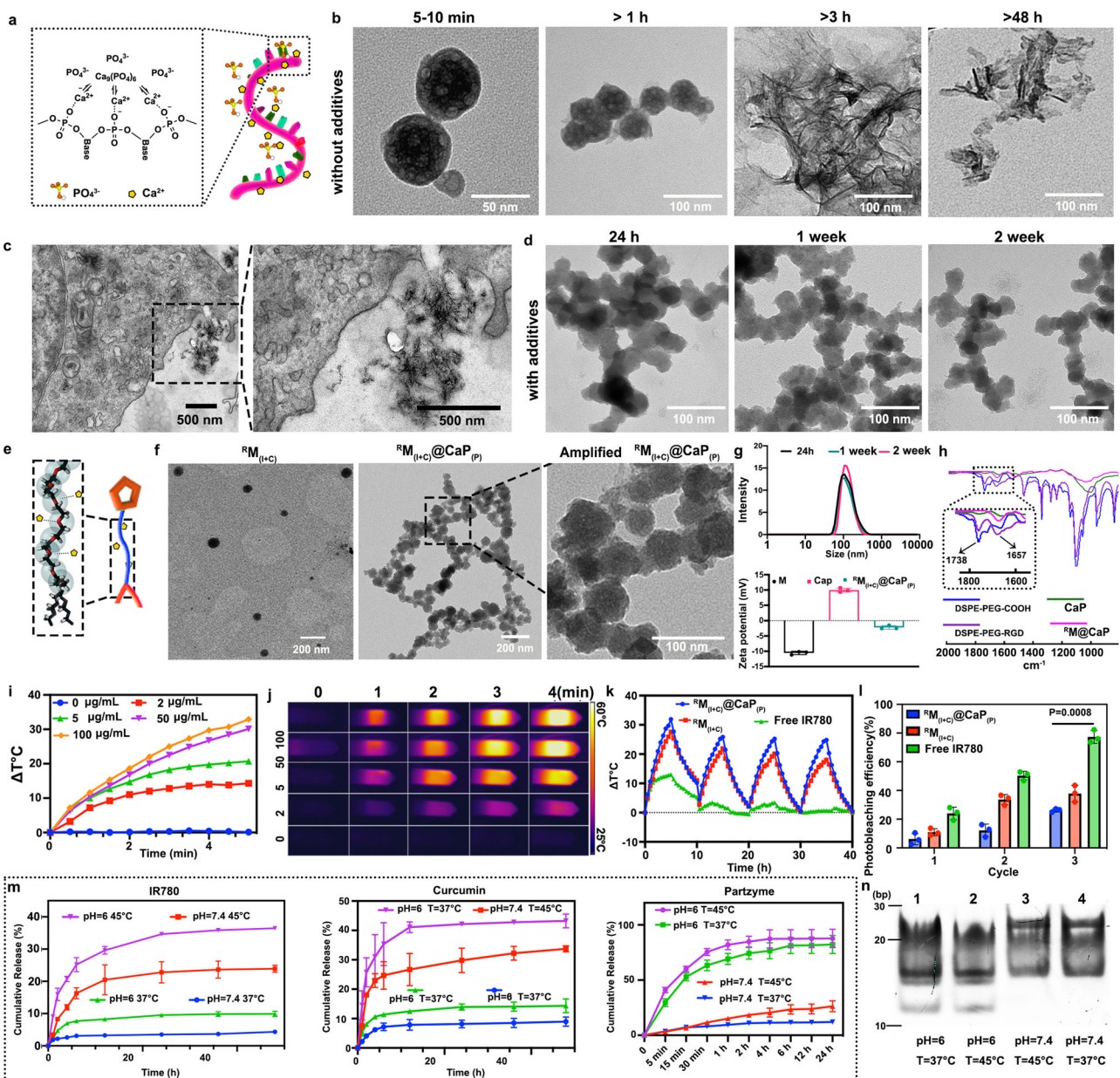

**Fig. 4 | Construction, drug loading and photothermal characterization of the designed nanosystem. a** Schematic illustration of the DNA and CaP biomineralization process. **b** Degradation process of CaP in Milli-Q aqueous solution. **c** Panc-1 cells were incubated with APC nanoparticles for 24 h. **d** Stability for Zn-substituted CaP in Milli-Q aqueous solution. **e** Interaction between PEG and $Ca^{2+}$ and $Zn^{2+}$. Carbon, oxygen, and hydrogen atoms are labeled with black, red, and white, respectively. **f** TEM images of $^RM_{(I+C)}$ and $^RM_{(I+C)}@CaP_{(p)}$. **g** Size distribution of $^RM_{(I+C)}@CaP_{(p)}$ NPs within 2 weeks and zeta potential results for $^RM_{(I+C)}$, Zn-doped CaP and $^RM_{(I+C)}@CaP_{(p)}$ ($n = 3$ independent experiments, and the data are presented as the mean values ± SDs). **h** FTIR results for Zn-substituted CaP and $^RM@CaP$.

concentrations of IR780. **j** The infrared thermal images correspond with photothermal heating curves. **k** Temperature elevation of $^RM_{(I+C)}@CaP_{(p)}$ NPs over four rounds of NIR on–off irradiation cycles. **l** Photobleaching effects of pure drug and nanoparticles after three rounds of laser irradiation ($n = 3$ independent experiments, and the data are presented as the mean values ± SDs). **m** Drug-releasing capacity of $^RM_{(I+C)}@CaP_{(p)}$ under different situations ($n = 3$ independent experiments and the data are presented as the mean values ± SDs). **n** Cleavage capacity of HSP70 mRNA after partzyme release from different environments. All statistics were calculated using two-tailed paired $t$ tests, and the experiments in (**b–d**, **f**, **n**) were repeated three times independently with similar results. The source data from (**g**, **h**, **l**, **k**, **i**, **m**) are provided as a Source Data file.

IR780 and 2 mg Cur within 50 mg DSPE-PEG-RGD) exhibited the best CDI = 0.52 (Tables S3 and S4).

After attaining the proper drug loading, we further determined the optimum dose of partzyme, and $PO_4^{2-}$, $Zn^{2+}$ and $Ca^{2+}$ were utilized to mineralize partzyme (P), as shown in Fig. 4f, $^RM_{(I+C)}@CaP_{(p)}$. For silencing miRNA-21 in pancreatic cancer cell lines, it has been reported that the concentration of miRNA-21 antisense oligonucleotides needs to reach 100 nM to achieve a significant gene-silencing effect[57,58].

Hence, the therapeutic concentration of partzyme, which also serves as a miRNA silencing agent, should also reach 100 nM. To confirm the efficient partzyme loading content of our designed nanosystem, we carried out a gradient partzyme loading test (Supplementary Fig. 2). Left and right partzymes were labeled with Cy3 and mixed with 1 ml $^RM_{(I+C)}$ solution (one/tenth upper-mentioned micelle volume, which contained 0.4 mg IR780 and 0.2 mg Cur) to yield different concentrations of each partzyme (1, 10, 50, 100, and 200 μM). After

calcium phosphate mineralization, we found that all partzymes were centrifuged to the bottom of the tube (even when the content was as high as 200 µM), and there was no leakage of partzyme after several washes with Milli-Q water. Therefore, the system had excellent partzyme encapsulation efficiency and eliminated the burst release phenomenon of oligonucleotides through a rational biomineralization process to easily achieve a 100 nM therapeutic concentration.

To further demonstrate the successful synthesis of $^RM_{(I+C)}$, $CaP_{(p)}$ and $^RM_{(I+C)}@CaP_{(p)}$, dynamic light scattering (DLS) and Fourier transform infrared spectroscopy (FTIR) were used to characterize the particle size, zeta potential and characteristic chemical bond of the nanocarrier. The particle size analysis revealed that $^RM_{(I+C)}$ possessed a similar average diameter as $^RM_{(I+C)}@CaP_{(p)}$, at ~100 ± 15 nm (Supplementary Fig. 3), which was larger than the TEM results (~50 nm) due to the aqueous layer out of the DLS specimen[59]. Notably, $^RM_{(I+C)}@CaP_{(p)}$ was found to be very stable (with the same size) in aqueous solution for more than 2 weeks (Fig. 4g), suggesting that this nanosystem possessed excellent stability, which may have occurred through a combination of zinc-additive and steric repulsion forces between the PEG chains. Additionally, in PBS buffer at pH = 7.4 (Fig. 4g), $^RM_{(I+C)}$ was observed to have a zeta potential of −10.21 mV, which may have been attributable to the carboxyl groups on RGD. Furthermore, the $Zn^{2+}$-substituted $CaP_{(p)}$ showed a strong positive charge (10.01 mV). Through the elemental mapping experiment, we found that the Zn ions and calcium ions were distributed throughout the NPs; hence, the positive charge may have been due to the uncoordinated calcium and zinc ions on its surface (Supplementary Fig. 4). Finally, we found that the final formulation $^RM_{(I+C)}@CaP_{(p)}$ exhibited −2.53 mV, which might have been caused by the outermost exposed RGD that renders the formulation slightly electronegative. Based on the aforementioned results, we conducted stability assessments on the final formulation of our nanomaterial in accordance with the guidelines provided by the International Council for Harmonisation of Technical Requirements for Pharmaceuticals for Human Use (ICH) Supplementary Table S5. We investigated the stability of the formulation at refrigerator temperature (4–8 °C) and under accelerated conditions at 40 °C, by following with the ICH guidelines, specifically ICH Q1A(R2) for long-term stability testing. The results of these stability studies indicated that there was no significant increase (p > 0.05) in particle size or polydispersity index (PDI) throughout the 3-month experimental period.

The FT-IR spectrum of DSPE-PEG-COOH reflected an absorbance band at 1738 cm$^{-1}$, corresponding to the −COOH groups[60]. After RGD modification, there was a newly formed band at 1657 cm$^{-1}$ belonging to the O=C-N-H group between PEG and RGD connections within micelles[60]. Subsequently, the micelles were coated by CaP, and the band at 1657 cm$^{-1}$ was also found in the $^RM_{(I+C)}@CaP$ group compared with the pure CaP group. These results together demonstrated the successful incorporation of CaP nanoparticles with the DSPE-PEG-RGD micelles (Fig. 4h).

IR780, as a PTT and PDT agent, suffers from poor aqueous solubility and high photobleaching properties and was protected inside micelles in this study for higher drug loading content and enhanced therapeutic effects. To examine the photothermal conversion properties of the final formulation, $^RM_{(I+C)}@CaP_{(p)}$, which contains different concentrations of IR780 (0, 2, 5, 50, 100 µg/ml), was irradiated with an 808 nm laser (1 W cm$^{-2}$) for 300 s (Fig. 4i). The corresponding infrared photos are shown in Fig. 4j. The results showed that the photothermal conversion properties of $^RM_{(I+C)}@CaP_{(p)}$ displayed a positive correlation with the NP concentration. The solution temperature increased by up to 20 °C when only 5 µg/ml IR780 NPs were irradiated for 5 min (Fig. 4i, j), while pure water did not increase the temperature.

Furthermore, the photothermal conversion efficiency ($\eta$) of pure IR780, $^RM_{(I+C)}$ and $^RM_{(I+C)}@CaP_{(p)}$ NPs was also studied through NIR heating-cooling cycles (1.0 W cm$^{-2}$, 808 nm) with 50 µg/ml IR780

(Fig. 4k). ($\eta$) was calculated by following Eqs. (5) and (6)[59]:

$$\eta = \frac{hs\Delta T_{max} - Q}{I(1 - 10^{-A^{808}})} \qquad (5)$$

Here, $\Delta T_{max}$ is the maximum temperature change value, while $Q$ is the solvent increased heat, $I$ is the power of the laser, $A^{808}$ is the absorbance of the NPs at a wavelength of 808 nm, and hs is defined by another formula:

$$hs = \frac{mc}{\tau} \qquad (6)$$

in which $m$ and $c$ are the specific heat capacity and mass of the solution, respectively, while $\tau$ is the slope of the fitted line of time to $-\ln\frac{\Delta T}{\Delta T_{max}}$.

The photothermal conversion efficiency of the pure drug was only 23.70% (Supplementary Fig. 5) due to the poor stability of IR780 in aqueous solution. In contrast, the $\eta$ of $^RM_{(I+C)}$ was converted up to 29.25% (Supplementary Fig. 6), which demonstrated that the micelle could prevent IR780 degradation. Surprisingly, the $Zn^{2+}$-containing dense CaP mineralization layer further improved the $\eta$ of IR780 (33.70%, Supplementary Fig. 7), indicating that the final formulation significantly improved the photothermal stability of IR780. Furthermore, the photobleaching efficiency was additionally evaluated for the abovementioned IR780, $^RM_{(I+C)}$ and $^RM_{(I+C)}@CaP_{(p)}$ groups (Fig. 4l). After three cycles of irradiation (1 W cm$^{-2}$, 5 min, 808 nm), free IR780 showed an ~80% decrease in absorbance at 780 nm, illustrating that free IR780 was rapidly degraded. Conversely, $^RM_{(I+C)}$ reflected a 41% decrease in absorbance, and $^RM_{(I+C)}@CaP_{(p)}$ showed only a 23% decrease after 3 cycles of irradiation. Hence, compared with $^RM_{(I+C)}$, $^RM_{(I+C)}@CaP_{(p)}$ realized an effective photothermal conversion ability of IR780.

To further estimate the pH-responsive drug release capacity, $^RM_{(I+C)}@CaP_{(p)}$ NPs (containing 0.4 mg of IR780, 0.2 mg of Cur and 200 µM partzyme) were subjected to release tests by dialysis membranes. Since IR780 generates heat under laser irradiation, it will promote the diffusion of drug molecules from the material, so different drug release environments with pH and temperature were designed (pH = 7, 37 °C; pH = 7, 45 °C; pH = 6, 37 °C; pH = 6, 45 °C). Therapeutic agents were observed through UV−visible spectroscopy at 780, 425 and 560 nm (Supplementary Fig. 8). We observed that at 45 °C, there was an increased release of the drug. Approximately 35% IR780 and 43% Cur were released at pH = 6 and 45 °C, whereas only 9% IR780 and 14% Cur were released at pH = 6 and 37 °C. A similar phenomenon was also found in the pH = 7 group. More than 20% IR780 and 30% curcumin were released at pH = 7 and 45 °C. However, less than 5% of IR780 and 10% of curcumin were released at pH = 7 at 37 °C.

Furthermore, we observed a pronounced pH-responsive release profile for partzymes since CaP possessed excellent pH-sensitive degradation performance, and more than 90% of the partzyme was released from the system after 4 h incubation at pH = 6 (Fig. 4m, partzyme). However, the degradation of CaP did not mediate significant release of Cur and IR780 at pH 6 compared to partzymes loaded in the CaP shell, reflecting that IR780 and Cur were loaded mainly inside the micelles. Subsequently, 1 µl of released medium from each group was further incubated with 1 µl of miRNA-21 (10 µM) and 2 µl of HSP70 mRNA (10 µM). We found that the released media in the pH = 6 groups efficiently cleaved HSP70 (Fig. 4n). Rapid release of partzymes ensured silencing of HSP70 before PTT treatment and prevented the previously mentioned hysteretic effect.

Given the successful design and construction of the dual-gene regulation MNAzyme system and the productive fabrication of the $^RM_{(I+C)}@CaP_{(p)}$ nanovector with high partzyme loading and excellent aqueous stability, we next closely investigated the biocompatibility of

the carrier system by testing the toxicity of the pure nanocarrier and laser to the normal human dermal fibroblast NFDH cell line (Supplementary Fig. 9). We found that pure nanovehicle $^R$M@CaP at a maximum concentration of 40.5 µg/ml (which could carry 6 µg/ml IR780 and 3 µg/ml curcumin) was not toxic to healthy cells. Hence, based on the results in Supplementary Table S3, at effective therapeutic concentrations for PANC-1 cells (IR780: 4 µg/ml, Cur: 2 µg/ml), pure carriers should have negligible side effects. In addition, NFDH cells were irradiated with an 808 nm laser at different laser intensities (0, 0.5, 1, 1.5 and 2 W cm$^{-2}$), and it was found that a pure laser did not inhibit the growth of healthy cells (Supplementary Fig. 9). These results suggest that the $^R$M@CaP system holds good biocompatibility and that the laser power we selected is safe for healthy cells.

Next, the cellular uptake of the pure drug, $M_{(I+C)}@CaP_{(p)}$ and $^RM_{(I+C)}@CaP_{(p)}$ groups by PANC-1 cells and the healthy control cell line NHDF (after 1 h and 8 h incubation) was observed by tracking the fluorescence of IR780 (red), Cy3-modified partzymes (green) and Cur (light blue) via confocal laser scanning microscopy (CLSM) and flow cytometry (Fig. 5a and Supplementary Figs. 10–12). The therapeutic concentrations were as follows: IR780: 4 µg/ml, Cur: 2 µg/ml and partzyme: 2 µM. In the case of PANC-1 cells, the results revealed that the uptake content of each therapeutic agent gradually increased with the extension of the incubation time (Supplementary Figs. 10 and 5a). After 8 h of incubation with the pure drug group, IR780 and Cur were pronounced diffused into PANC-1 cells, while Cy3-labeled partzyme exhibited no internalization. However, for the $M_{(I+C)}@CaP_{(p)}$ group, IR780, Cur and Cy3-labeled partzyme were all endocytosed, which proved that the nanocarrier could ensure the internalization of partzyme. Meanwhile, after the decoration of RGD on top of the nanosystem, the $^RM_{(I+C)}@CaP_{(p)}$ group exhibited 2.2 times higher uptake efficiency than the $M_{(I+C)}@CaP_{(p)}$ group, which reflected that RGD, as a tumor-targeting tripeptide, could promote the endocytosis of NPs by pancreatic cancer PANC-1 cells. Inductively coupled plasma–optical emission spectrometry (ICP–OES) results indicated that the calcium ion concentration in the cancer cells was ~240 parts per million (ppm), which was sufficient to activate MNAzymes (Supplementary Fig. 10).

In contrast, for NHDF cells, the pure drug, $M_{(I+C)}@CaP_{(p)}$ and $^RM_{(I+C)}@CaP_{(p)}$ groups all showed negligible uptake after 1 h of treatment (Supplementary Fig. 11). Meanwhile, only sporadic drug fluorescence signals were exhibited for the $M_{(I+C)}@CaP_{(p)}$ and $^RM_{(I+C)}@CaP_{(p)}$ groups after 8 h of incubation (Supplementary Fig. 12). These phenomena may be attributable to the weaker uptake capacities of normal cells[41]. The surface modification of RGD did not promote the ingestion of NPs by healthy NHDFs.

Having determined that the $^RM_{(I+C)}@CaP_{(p)}$ group could effectively deliver multiple cargoes into PANC-1 cells, we explored the intracellular endosomal escape. For both groups, after NPs and cells were cultured together for 2 h and 24 h, the cell media were replaced, and a 1 W laser was applied for 5 min. Here, IR780 is presented in red, and the lysosome is labeled in green. Pearson correlation coefficients (PCCs) were utilized to determine whether the two colors colocalized (>0.5 indicates moderate correlation; <0.5 indicates low correlation)[61,62]. After 2 h of treatment with $^RM_{(I+C)}@CaP_{(p)}$ NPs without laser irradiation, we found that the red fluorescence (IR780) overlapped well with the green fluorescence of lysosomes (Fig. 5b). In addition, the PCC was calculated as 0.673 (higher than 0.5), demonstrating the colocalization of IR780 and lysosomes. After 1 W cm$^{-2}$ 808 nm laser irradiation for 5 min, the PCC number dropped to 0.582 but still showed a colocalized effect. For the $^RM_{(I+C)}@CaP_{(p)}$ NPs after 24 h of treatment, lower overlap between IR780 and lysosomes was observed, and the PCC value was 0.454. Notably, there was almost no overlap between red and green fluorescence after laser irradiation (PCC = 0.413), confirming that the therapeutic agents could efficiently escape from the lysosome.

After validating that the therapeutics could be efficiently delivered to the cytoplasm, we further studied the functional performance of each nanosystem component (MNAzyme, IR780, Cur and Ca$^{2+}$) in PANC-1 cells. For the MNAzyme system, self-assembly into MNAzyme should be achieved by the released partzymes and endogenously overexpressed miRNA-21, which then autocatalytically trigger its substrate (HSP70 mRNA) cleavage function with the assistance of Ca$^{2+}$ (Fig. 5c). Subsequently, the silencing of miRNA-21 can mediate the increased expression of the tumor suppressor protein PTEN, while the HSP70 protein should be downregulated after HSP70 mRNA cleavage.

Based on the above assumptions, to verify that the partzymes could bind to intracellular miRNA-21, the fluorescence in situ hybridization (FISH) method was used to observe intracellular free miRNA-21 (Fig. 5d). Four groups, including the PBS, IR780+Cur, $^RM_{(I+C)}@CaP$ (without partzymes) and $^RM_{(I+C)}@CaP_{(p)}$ (with partzymes) groups, were used, and endogenous miRNA-21 was labeled red. From the results, we observed that the PBS groups exhibited tremendous Cy5.5 fluorescence (miRNA), while when IR780 and curcumin were added, the fluorescence intensity of miRNA decreased by nearly 32%. It has been reported that curcumin can regulate a variety of miRNAs (including miRNA-21) in other cancer cell lines[63]. Hence, curcumin could also inhibit endogenous miRNA-21 in the PANC-1 cell line. Of note, during the experiments, all the therapeutics containing cell culture medium were replaced at the 6 h timepoint, which resulted in a shorter intracellular retention time for the pure drug, while the nanosystem promoted the retention time and therapeutic efficiency of Cur. As a result, the red fluorescence intensity in the $^RM_{(I+C)}@CaP$ group was decreased by ~76%. More interestingly, from the results of the $^RM_{(I+C)}@CaP_{(p)}$ group, the red fluorescence intensity was reduced almost to zero, indicating that the partzyme system could efficiently utilize miRNA-21 and ensure MNAzyme formation.

Subsequently, to explore whether MNAzyme can effectively downregulate the expression of HSP70 protein and mediate the increase in the expression of PTEN protein in PANC-1 cells, western blot experiments were performed, and the relative HSP70/GAPDH and PTEN/GAPDH ratios were calculated with ImageJ, as shown in Fig. 5e. For the HSP70 protein, compared with | PBS (1 and 2), we found that IR780+Cur with laser treatment (4) significantly upregulated HSP70 expression (1.8 times), which indicates that PTT can activate HSP70 to protect cells from thermal stress[64]. Surprisingly, for the $^RM_{(I+C)}@CaP$ groups (5 and 6), the expression of HSP70 was downregulated by nearly 40% (-laser) and 20% (+laser). It has been reported that the generation of HSPs highly depends on ATP production[8,9]. Since our nanosystem will trigger multichannel Ca$^{2+}$ ions inside cells to cause mitochondrial calcium overload (Figs. 6 and 7), ATP production by damaged mitochondria should decline and lead to a decrease in HSP70 content. Ultimately, from the results of the $^RM_{(I+C)}@CaP_{(p)}$ groups with partzymes 7 and 8, the HSP70 gene was fully eliminated for both the ±laser groups, suggesting the effective HSP70 mRNA cleavage function of the MNAzyme system.

PTEN protein expression should be upregulated owing to the silencing of miRNA-21. From the results, compared with the PBS groups (1 and 2), the pure drug groups (3 and 4) showed slight upregulation of PTEN expression due to the inhibitory effect of curcumin. Meanwhile, the relative ratio of PTEN/GAPDH for the $^RM_{(I+C)}@CaP$ groups (5 and 6) was ~2 times higher than that of the PBS groups, which might have been caused by the long-term Cur-mediated decrease in miRNA-21. Furthermore, for the $^RM_{(I+C)}@CaP_{(p)}$ groups (7 and 8), a significant increase in the PTEN/GAPDH relative ratio (3 times higher than that of the PBS groups) was observed, demonstrating the miRNA-21 silencing effect of the MNAzyme system.

Considering that the MNAzyme system has a targeted therapeutic effect on cancer cells, we further investigated MNAzyme function in healthy NFDH cells in eight groups. The FISH method was first utilized to compare the miRNA-21 content between NFDH cells and PANC-1

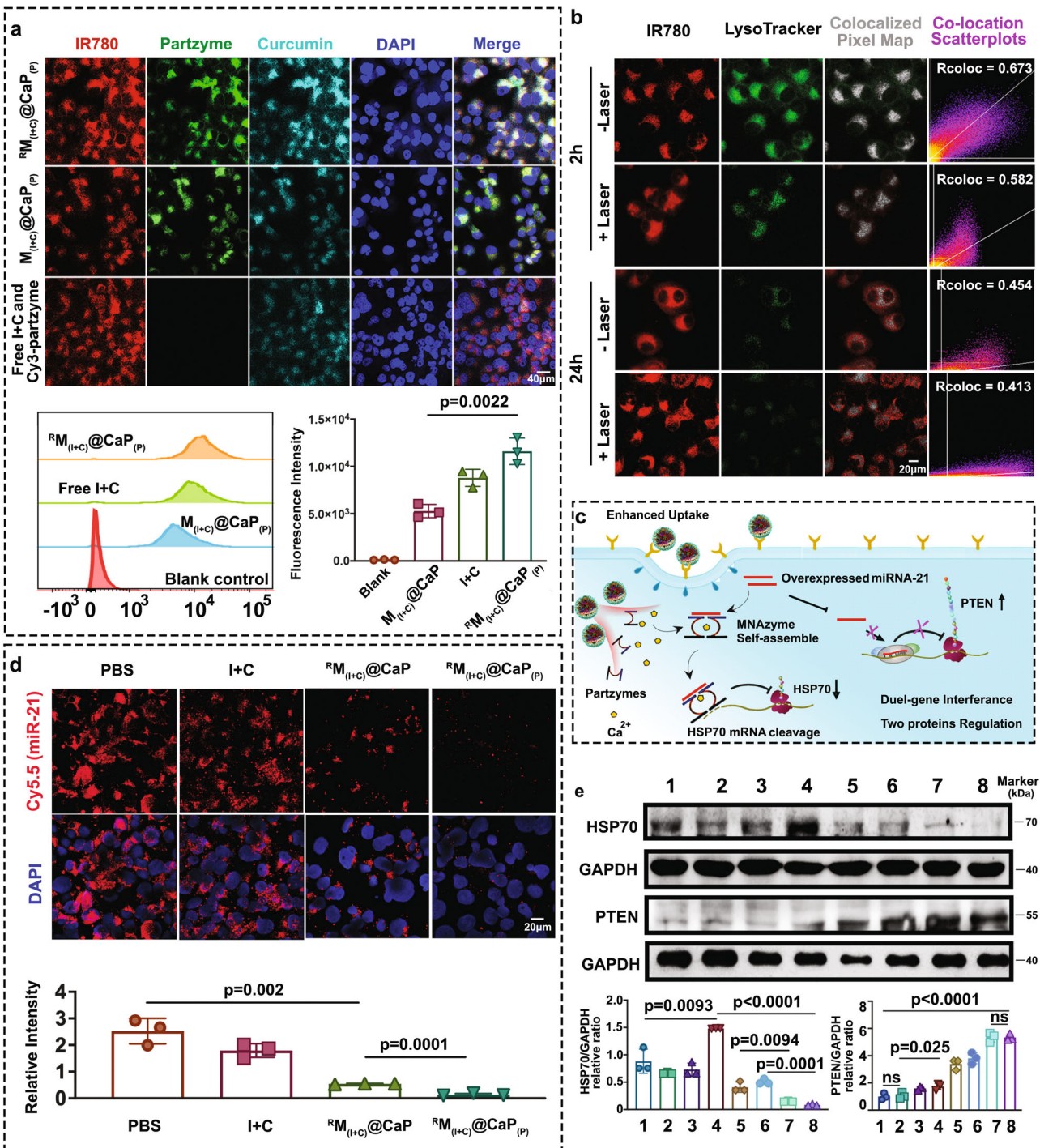

**Fig. 5 | Therapeutic delivery by nanosystems and the intracellular function of the MNAzyme. a** Uptake of pure drug, $M_{(I+C)}@CaP_{(p)}$ and $^RM_{(I+C)}@CaP_{(p)}$ groups by PANC-1 cells ($n$ = 3 independent experiments, and the data are presented as the mean values ± SDs). **b** Lysosomal escape profiles of $^RM_{(I+C)}@CaP_{(p)}$ groups (±Laser) by PANC-1 cells at 2 h and 24 h. The Pearson value Rcoloc was calculated by using ImageJ with the Coloc 2 plugin. **c** The schematic diagram depicts the intracellular self-assembly of the MNAzyme system for regulating the protein expression of PTEN and HSP70. **d** Intracellular miRNA-21 content was determined by the fluorescence in situ hybridization (FISH) method ($n$ = 3 independent experiments and

the data are presented as the mean values ± SDs). **e** The expression of HSP70 and PTEN protein was evaluated by western blotting. The final concentrations of the components in the different groups were as follows: IR780: 4 μg/ml, Cur: 2 μg/ml, each partzyme: 2 μM) ($n$ = 3 independent experiments, and the data are presented as the mean values ± SDs). Groups 1–8 represent the following: 1: PBS, 2: PBS +Laser, 3: I + C, 4: I + C +Laser, 5: $^RM_{(I+C)}@CaP$, 6: $^RM_{(I+C)}@CaP$ +Laser, 7: $^RM_{(I+C)}@CaP_{(p)}$, and 8: $^RM_{(I+C)}@CaP_{(p)}$ +Laser. All statistics were calculated using two-tailed paired $t$ tests, and the experiments in (**a**, **b**, **d**, **e**) were repeated three times independently with similar results. The source data from (**a**, **d**, **e**) are provided as a Source Data file.

cells (Supplementary Fig. 13). The green fluorescence in NFDH cells was clearly ~7 times weaker than that in PANC-1 cells, indicating that partzymes might not effectively use miRNA-21 to construct the MNA-zyme system. Moreover, due to the absence of RGD receptors on the

surfaces of healthy cells, the construction of the MNAzyme system was hampered by both poor partzyme delivery efficiency and the absence of miRNA-21, which could not further regulate the expression of HSP70 proteins. (Supplementary Fig. 14). Western blotting was then applied

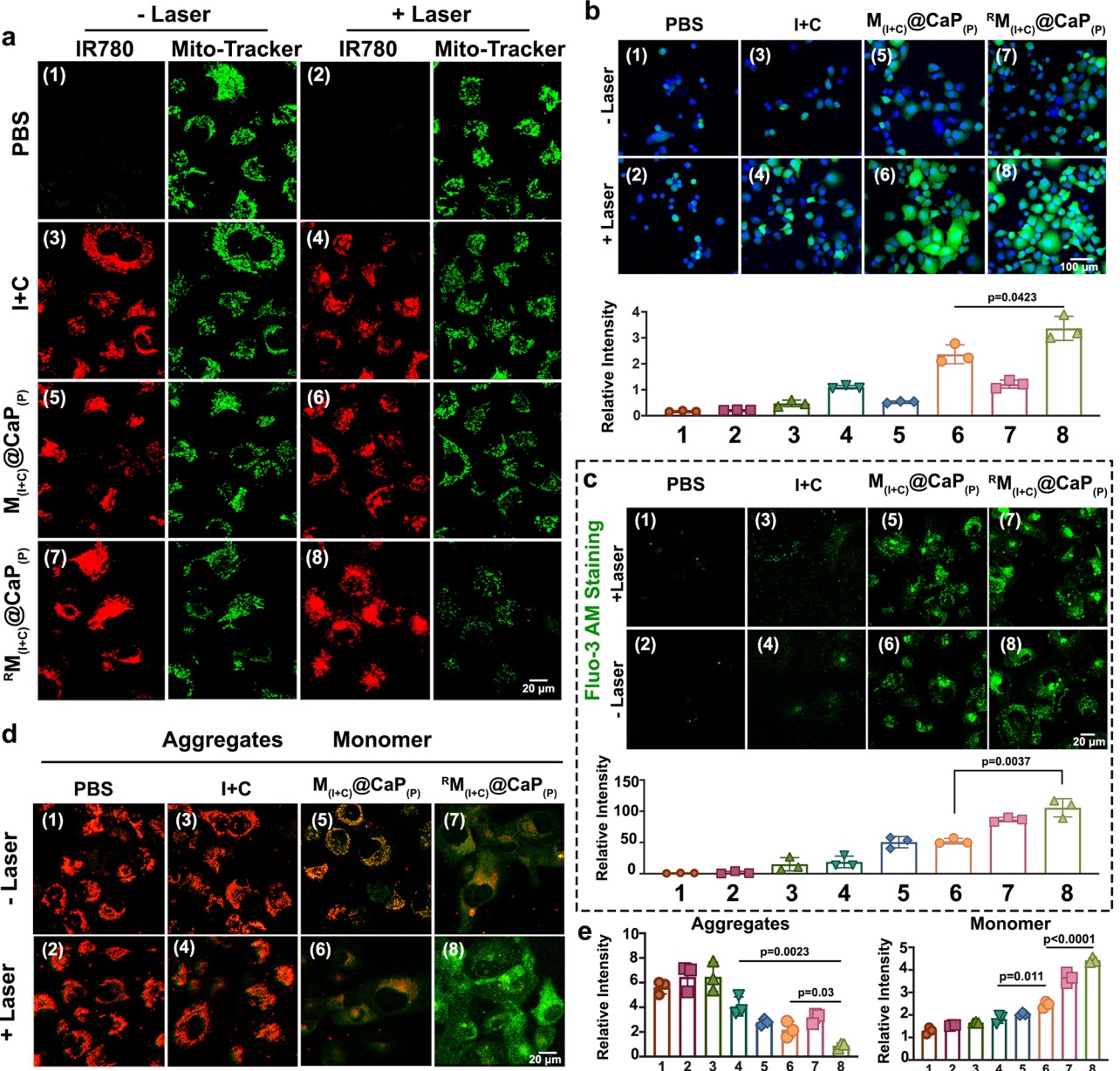

**Fig. 6 | Multidimensional mitochondrial therapeutics mediated by IR780, Cur and Ca²⁺. a** Colocalization between IR780 and mitochondria. **b** $^1O_2$ generation capability of each treatment group ($n = 3$ independent experiments, and the data are presented as the mean values ± SDs). **c** Ca²⁺ concentration level visualization by using Fluo-3 AM as a labeling dye ($n = 3$ independent experiments and the data are presented as the mean values ± SDs). **d** ΔΨm detection by tracking of the JC-1 monomer and aggregates through CLSM. **e** Relative fluorescence intensity of JC-1 signals for each group as calculated by ImageJ. The eight groups are labeled from 1 to 8, including PBS (1), PBS +Laser (2), IR780+Cur (3), IR780+Cur +Laser (4), $M_{(I+C)}@CaP_{(P)}$ (5), $M_{(I+C)}@CaP_{(p)}$ +Laser (6), $^RM_{(I+C)}@CaP_{(p)}$ (7) and $^RM_{(I+C)}@CaP_{(p)}$ +Laser (8). The final concentrations of the components in the different groups were as follows: IR780: 4 µg/ml, Cur: 2 µg/ml, and each partzyme: 2 µM ($n = 3$ independent experiments, and the data are presented as the mean values ± SDs). All statistics were calculated using two-tailed paired $t$ tests, and the experiments in (**a**–**d**) were repeated three times independently with similar results. The source data from (**b**, **c**, **e**) are provided as a Source Data file.

to test the generation of HSP70 in NFDH cells. The results showed that IR780 + Cur + laser (4) increased HSP70 expression by 2 times compared with that in the PBS group, while $^RM_{(I+C)}@CaP$ + laser (6) and $^RM_{(I+C)}@CaP_{(p)}$ + laser (8) increased the HSP70/GAPDH relative ratio by 3.4 times (Supplementary Fig. 14), demonstrating that the nanosystem did not affect the thermal protection mechanism of healthy cells.

After confirming the cancer-specific dual-gene regulatory capability of the MNAzyme system, we further studied subcellular-based ion therapy with our nanoplatform in PANC-1 cells. Multidimensional mitochondrial therapeutics was realized through the following synergistic mechanisms: (1) IR780 targeted mitochondria to generate PTT

and PDT therapy; (2) CaP-composed nanocarriers released tremendous amounts of Ca²⁺, mediating intramitochondrial Ca²⁺ overload; and (3) Cur further promoted the release of Ca²⁺ from the ER and inhibited Ca²⁺ efflux pumps in cancer cells.

We first verified the mitochondria-targeted therapy with IR780 through CLSM, as shown in Fig. 6a. Eight groups, including PBS ±Laser, I + C ±Laser, $M_{(I+C)}@CaP_{(p)}$ (without RGD) ±Laser and $^RM_{(I+C)}@CaP_{(p)}$ (with RGD) ±Laser, were established and marked as 1 to 8. The laser groups were under $1\,W\,cm^{-2}$ 808 nm irradiation for 5 min. IR780 exhibited red fluorescence, and mitochondria were marked in green. As expected, IR780 colocalized with mitochondria in each treatment

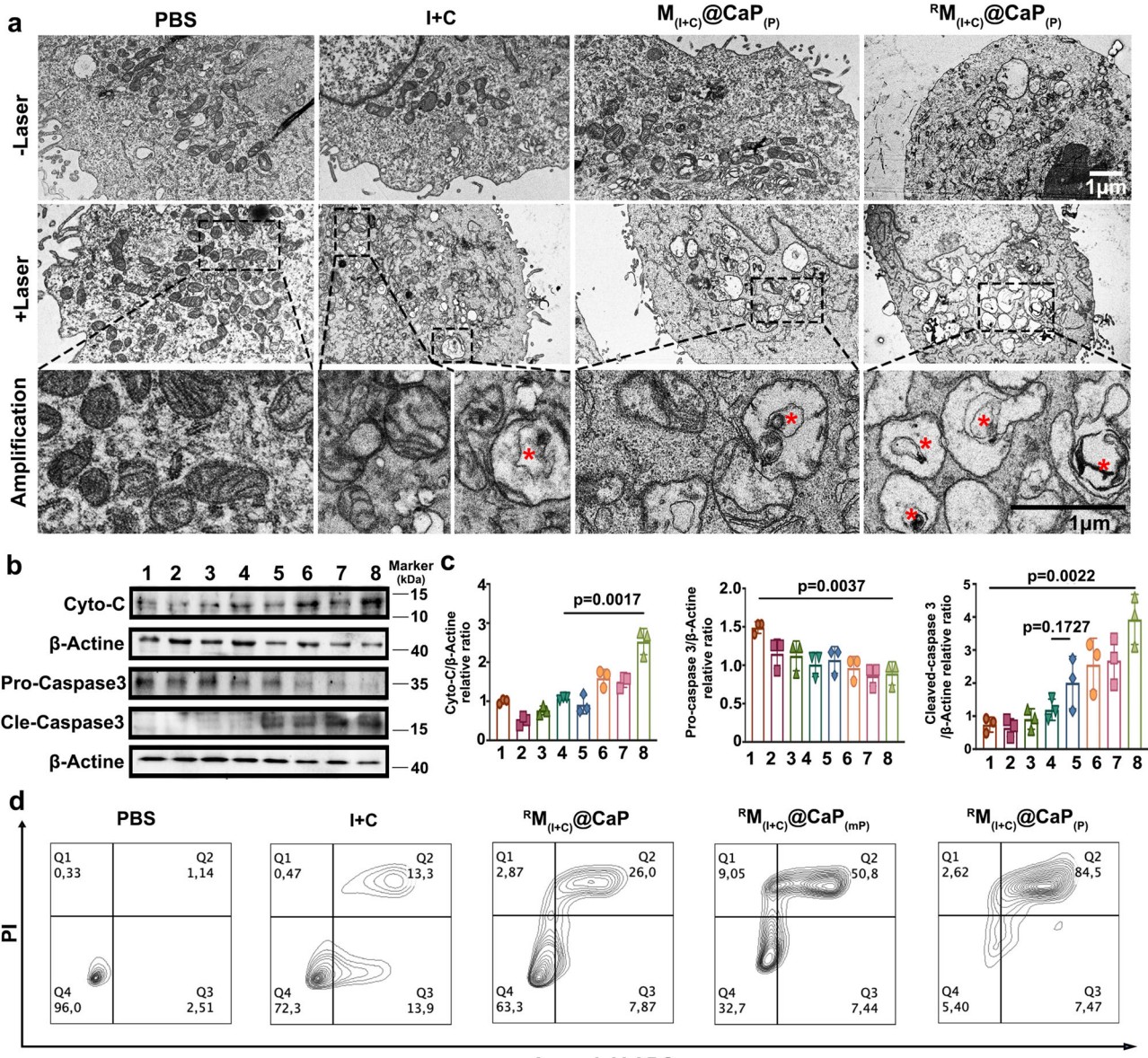

**Fig. 7 | Mitochondrial morphology changes and the apoptosis mechanism.**
**a** Bio-TEM image for mitochondrial morphology observation. **b** Western blot experiments for apoptosis-related protein detection. Eight groups (1–8) were tested, including the PBS, PBS +Laser, I + C, I + C +Laser, $^R$M$_{(I+C)}$@CaP, $^R$M$_{(I+C)}$@CaP +Laser, $^R$M$_{(I+C)}$@CaP$_{(p)}$ and $^R$M$_{(I+C)}$@CaP$_{(p)}$ +Laser groups. (*n* = 3 independent experiments). **c** The relative ratio of each protein was calculated with ImageJ (*n* = 3 independent experiments, and the data are presented as the mean values ± SDs).

**d** Apoptosis analysis of PANC-1 cells under Annexin V-APC/PI staining. (mP means mutated partzyme, which can only target miRNA but cannot achieve HSP70 silence; all groups were under laser irradiation). All statistics were calculated using two-tailed paired *t* tests, and all experiments (**a**, **b**, **d**) were repeated three times independently with similar results. The source data from (**b**, **c**) are provided as a Source Data file.

group. Compared with those of the PBS and pure drug groups (groups 1–4), the green fluorescence signals of the M$_{(I+C)}$@CaP$_{(p)}$ group without laser (5) were decreased, which might have been caused by Ca$^{2+}$-mediated mitochondria disruption. M$_{(I+C)}$@CaP$_{(p)}$ +Laser (6) showed less green signal than (5) due to the synergistic effect of IR780 and calcium overload. In contrast, after modification with RGD, the $^R$M$_{(I+C)}$@CaP$_{(p)}$ ±Laser group (7 and 8) exhibited the least green fluorescence, indicating the most severe mitochondrial damage caused by RGD-mediated enhanced endocytosis of the nanosystem by cancer cells.

In addition, the intracellular $^1$O$_2$ generation capability was also monitored to investigate the PDT effect of IR780. The same groups mentioned above were implemented, and the ROS were tracked with 2′,7′-dichlorofluorescein diacetate (which emits a green signal after

oxidation by $^1$O$_2$), as depicted in Fig. 6b. All +Laser treatment groups produced observable $^1$O$_2$ upon laser irradiation for 5 min (808 nm 1.0 W cm$^{-2}$). We found that the pure drug I + C +Laser group (4) showed less relative fluorescence intensity, which was due to poor drug stabilization and solubility in the cell culture medium. The $^R$M$_{(I+C)}$@CaP$_{(p)}$ +Laser group (8) produced the strongest green signal; comparatively, only 75% of the relative fluorescence intensity in the M$_{(I+C)}$@CaP$_{(p)}$ +Laser group (6) was found, which was calculated by ImageJ, reflecting the enhanced therapeutic effect mediated by RGD decoration.

Later, Ca$^{2+}$-based ion therapy was further studied by using Fluo-3 AM (green fluorescence) to visualize intracellular Ca$^{2+}$ via CLSM, and the relative intensity of Ca$^{2+}$ was calculated with ImageJ (Fig. 6c). The same groups were selected, and an 808 nm 1 W cm$^{-2}$ laser was chosen for 5 min. As expected, a noticeable increase in Ca$^{2+}$ concentration was

found in the $^R$M$_{(I+C)}$@CaP$_{(p)}$ ±Laser groups (7 and 8), indicating the high efficiency of our designed nanoplatform for inducing mitochondrial Ca$^{2+}$ overload. Meanwhile, compared with M$_{(I+C)}$@CaP$_{(p)}$ ±Laser (5 and 6), Groups 7 and 8 showed 2 times higher Ca$^{2+}$ relative intensity levels, and the increases were attributable to the RGD-mediated enhancement of the endocytosis effect. In addition, with the assistance of curcumin, the $^R$M$_{(I+C)}$@CaP$_{(p)}$ group showed an excellent plasma membrane calcium pump (PMCA ATPase) silencing effect and achieved higher Ca$^{2+}$ levels than the group without Cur (Supplementary Fig. 15), indicating the synergistic function of curcumin and Ca$^{2+}$-containing nanocarriers.

Mitochondrial transmembrane potential (ΔΨm) represents the hyperpolarization/depolarization of mitochondria. To ascertain that the mitochondria experienced a disruption effect from our designed nanosystem, JC-1 dye was further employed as a sensor (JC-1 aggregates exhibit a red signal for healthy cells with high ΔΨm, and the JC-1 monomer exhibits a green signal for damaged cells with low ΔΨm). The same groups were used as elucidated in Fig. 6d, and the relative signal intensities of aggregates and monomers were calculated by ImageJ (Fig. 6e). After laser irradiation for 5 min (808 nm 1.0 W cm$^{-2}$), compared with the PBS group (1 and 2), the I + C +Laser group (4) showed 1.5 times higher green fluorescence, suggesting a decrease in ΔΨm. In contrast, the 2-fold higher monomer content in the M$_{(I+C)}$@CaP$_{(p)}$ +Laser group (6) suggested synergistic mitochondrial damage by IR780 and Ca$^{2+}$ overload. Surprisingly, with RGD decoration, the $^R$M$_{(I+C)}$@CaP$_{(p)}$ ±Laser group (7 and 8) showed 3.16 and 3.6 times higher monomer signals, respectively, indicating that RGD modification significantly enhanced the therapeutic efficiency of the pure drug and efficiently mitigated mitochondrial dysfunction by inducing the synergistic effect of PDT/PTT and Ca$^{2+}$ overload.

Later, to directly observe the mitochondrial morphology changes, bio-TEM was used to analyze each aforesaid group, and the inclusions inside mitochondria were marked with a red star (Fig. 7a). In general, a higher energy requirement for cancer cells results in more mitochondria with abundant cristae. As expected, we found tremendous amounts of mitochondria with dense cristae in PANC-1 cells in the PBS ± Laser groups. In addition, mild mitochondrial destruction (slight swelling) was observed in the I + C +Laser groups, while the therapeutic effect was not prominent. Of note, noticeable mitochondrial swelling and cavitation effects were detected in the M$_{(I+C)}$@CaP$_{(p)}$ +Laser groups, which proved that the nanosystem could significantly enhance mitochondrial dysfunction through IR780 and superabundant calcium. It is worth mentioning that with RGD modification, negligible cristae were found in the $^R$M$_{(I+C)}$@CaP$_{(p)}$ group, suggesting that RGD mediated enhanced curative efficiency.

MNAzyme-mediated enhanced photothermal therapy and Ca$^{2+}$ overload treatment increased mitochondrial permeability, resulting in the release of Cyto-C and finally activating Caspase-3 to mediate apoptosis. The apoptosis mechanism was investigated through western blot experiments (Fig. 7b), and the relative ratio of each protein was calculated by ImageJ (Fig. 7c). From the results, compared with the PBS group (1 and 2), the I + C +Laser group (4) showed a slight increase in Cyto-C, which reflected inefficient damage to mitochondria. In contrast, the $^R$M$_{(I+C)}$@CaP +Laser group (6) showed ~1.5 times higher Cyto-C release than the I + C +Laser group (4), which was related to enhanced mitochondrial calcium overload. Importantly, due to the MNAzyme-mediated PTT-enhancing effect, the $^R$M$_{(I+C)}$@CaP$_{(p)}$ +Laser treatment (8) greatly led to the release of Cyto-C from mitochondria into the cytoplasm (2.5 times higher than that in the PBS group), which mediated pro-caspase cleavage, and cleaved caspase-3 was found at up to 3 times higher levels than that in the PBS group, as shown in Fig. 7c. Therefore, the $^R$M$_{(I+C)}$@CaP$_{(p)}$ nanosystem efficiently upregulated apoptosis-related proteins, resulting in significant apoptosis.

Based on the aforementioned experiments, we successfully validated the upregulation effect of the nanosystem on PTEN, the silencing effect on HSP70, and the calcification effect on mitochondria. Given ample evidence that these rationally designed nanoformulations can achieve excellent cancer therapy, we further investigated their multiple synergistic effects through WST-1 and flow cytometry for both the NHDF and PANC-1 cell lines (Supplementary Figs. 16–17 and 7d). Different groups, including PBS, I + C, $^R$M$_{(I+C)}$@CaP (without partzymes), $^R$M$_{(I+C)}$@CaP (with mutated partzymes) and $^R$M$_{(I+C)}$@CaP$_{(p)}$ (with partzymes), were implemented under laser irradiation at 808 nm and 1 W cm$^{-2}$ for 5 min. The mutated partzyme can only bind with miRNA-21 but cannot recognize HSP70. The final concentrations in the different groups were IR780: 4 μg/ml, Cur: 2 μg/ml, and partzyme: 2 μM. After 24 h of treatment, compared with the PBS group, the pure drug I + C group showed ~20% inhibition of healthy NHDFs. However, the $^R$M$_{(I+C)}$@CaP, $^R$M$_{(I+C)}$@CaP$_{(mP)}$ and $^R$M$_{(I+C)}$@CaP$_{(p)}$ groups exhibited almost no killing effect on healthy NHDFs due to less NP endocytosis and the miRNA-21 targeting effect of the MNAzyme system (Supplementary Fig. 16). For PANC-1 cancer cells, compared with the PBS group, the pure drug I + C group exhibited an ~48% inhibitory effect. However, 63% of the cancer cells were inhibited in the $^R$M$_{(I+C)}$@CaP group, which suggested that Ca$^{2+}$ enhanced the mitochondrial dysfunction effect. Approximately 75% of cells were inhibited in the $^R$M$_{(I+C)}$@CaP$_{(mP)}$ group (Supplementary Fig. 17), indicating that the MNAzyme system boosted cancer therapy through miRNA-21 silencing. Of note, almost 90% of the cancer cells were inhibited in the $^R$M$_{(I+C)}$@CaP$_{(p)}$ group, which revealed that the partzyme could achieve both miRNA and HSP70 regulation and realize an excellent synergistic effect.

From the apoptosis assay results, within 8 h of treatment (IR780: 2 μg/ml, Cur: 1 μg/ml, partzyme: 1 μM), ~13.3% of the cells in the pure drug I + C group had entered the late apoptotic phase, while the percentage was 26.0% for the $^R$M$_{(I+C)}$@CaP group, reflecting Ca$^{2+}$-mediated synergistic ion therapy. Moreover, 50.8% of the cells in the $^R$M$_{(I+C)}$@CaP$_{(mP)}$ group were shown to enter the late apoptotic state (Fig. 7d), which indicates the existence of a synergistic effect through mutated partzyme-based miRNA regulation. Finally, for the final formulation, 84.5% of the cells entered the late apoptotic phase, which was attributed to the MNAzyme-enhanced PTT, PTEN regulation, PDT and Ca$^{2+}$ overload. The gating/sorting strategies are shown in Supplementary Fig. 18.

Encouraged by the selective suppressing ability and excellent biocompatibility in vitro, we further explored the performance of each nanosystem in vivo, which was approved by the Institutional Animal Care and Use Committee, Zhejiang Center of Laboratory Animals (Approval No. ZJCLA-IACUC-20020090). A pancreatic orthotopic tumor model was constructed by injecting 10$^7$ PANC-1-luc cells into the pancreatic site. The blood circulation and the blood half-life of the nanomachine were investigated first to determine its pharmacokinetics and potential efficacy in vivo (Supplementary Fig. 19). Three groups, including the pure drug (IR780 + curcumin + partzymes), M$_{(I+C)}$@CaP$_{(p)}$ and $^R$M$_{(I+C)}$@CaP$_{(p)}$ groups, were compared. From the results, we found that the M$_{(I+C)}$@CaP$_{(p)}$ and $^R$M$_{(I+C)}$@CaP$_{(p)}$ groups had longer circulation times, with elimination half-lives ($t_{1/2β}$) of 26.69 h and 25.53 h, respectively, compared to the free drug, which had a $t_{1/2β}$ of 5.34 h. The prolonged circulation time of NPs is thought to be due to the PEG shell, which shields the nanoparticle from being recognized and cleared by the immune system.

Live animal imaging was then performed to demonstrate the biodistribution of NPs after intravenous administration (Fig. 8a). Three groups, including the pure drug (IR780 + curcumin + partzymes), M$_{(I+C)}$@CaP$_{(p)}$ and $^R$M$_{(I+C)}$@CaP$_{(p)}$ groups, were tracked at 1, 8, 24 and 48 h timepoints (IR780: 1 mg/kg). For the pure drug I + C + P group, the majority of the signal was found throughout the mouse body at the 1 h timepoint, while a negligible signal was observed after 8 h due to the rapid metabolism of IR780. In contrast, M$_{(I+C)}$@CaP$_{(p)}$ NPs gradually released the drug, which reached its maximum concentration at the

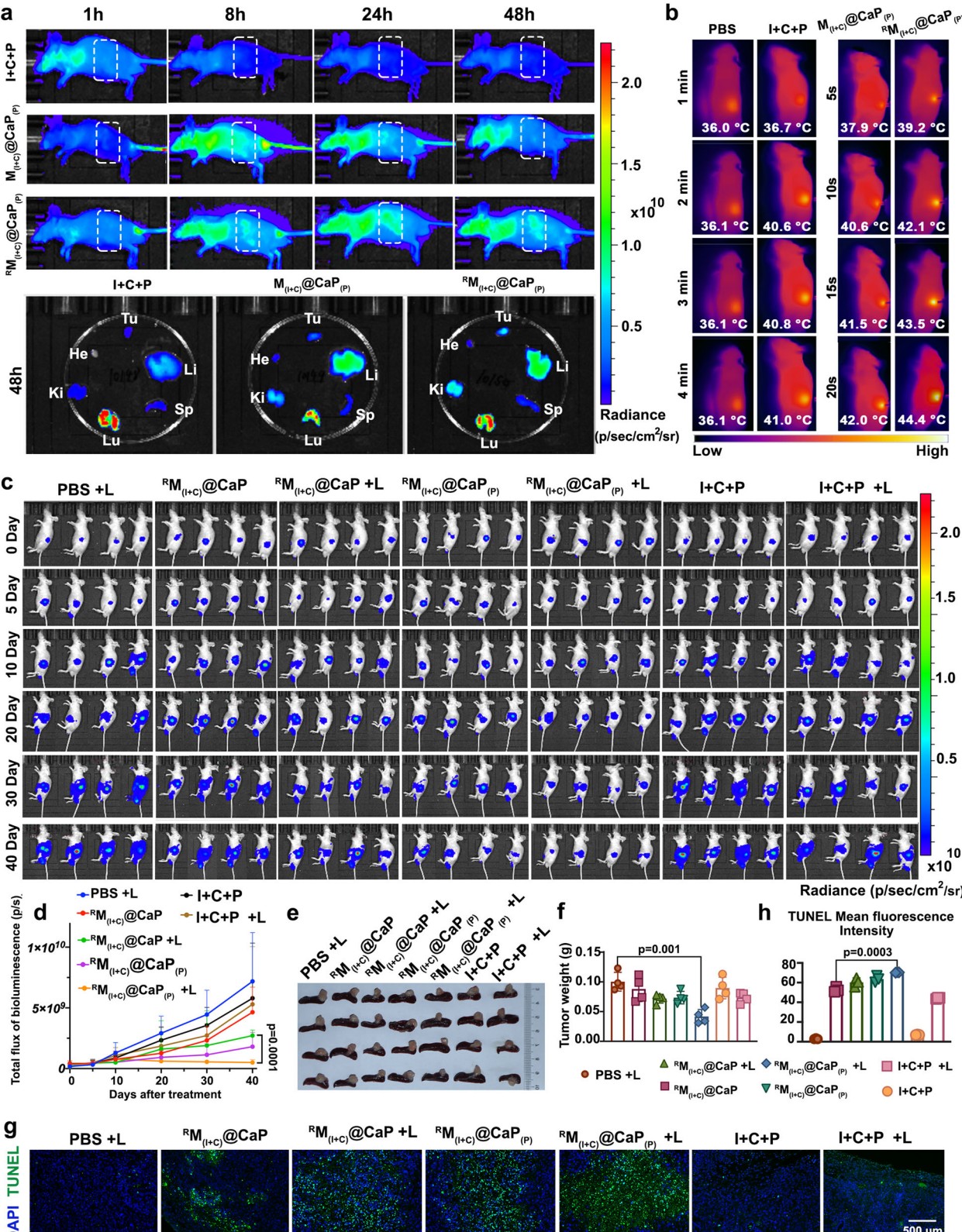

**Fig. 8 | Distributions and curative effects of different preparations in vivo. a** In vivo biodistribution of nanomaterial. Tu Tumor, Li Liver, Sp Spleen, Lu Lung, Ki Kidney, He Heart. **b** Tumor temperature elevation pictures. **c** Bioluminescence at different timepoints. **d** Quantitative analysis of total flux signal intensity (photons/sec) over time ($n = 4$ independent experiments, and the data are presented as the mean values ± SDs). **e** Isolated tumor photos. **f** Tumor weight ($n = 4$ independent experiments, and the data are presented as the mean values ± SDs). **g** TUNEL labeling of each tumor slice. **h** TUNEL fluorescence quantitated by ImageJ ($n = 3$ independent experiments, and the data are presented as the mean values ± SDs). All statistics were calculated using two-tailed paired $t$ tests, and all experiments were repeated three times independently with similar results. The source data from (**d**, **f**, **h**) are provided as a Source Data file.

tumor site after 8 h of intravenous administration. Moreover, due to RGD modification, the $^RM_{(I+C)}$@CaP$_{(p)}$ group showed better tumor accumulation, achieving the highest fluorescence signal among the groups. Subsequently, the main organs were further harvested and imaged to evaluate the drug biodistribution. Compared with the free drug and M$_{(I+C)}$@CaP$_{(p)}$ NPs, $^RM_{(I+C)}$@CaP$_{(p)}$ NPs resulted in pronounced fluorescence in the tumor tissue, proving that RGD can facilitate tumor targeting.

Later, to assess mild PTT efficiency, a tumor temperature elevation experiment was performed. Four groups (PBS, I + C + P, M$_{(I+C)}$@CaP$_{(p)}$, and $^RM_{(I+C)}$@CaP$_{(p)}$) were treated with a 1 W cm$^{-2}$ 808 nm laser at 24 h after drug administration (IR780: 1 mg/kg), and the results are elucidated in Fig. 8b. From the results, the PBS group elicited the least change in temperature even after 4 min of irradiation. The pure drug I + C + P group showed an ineffective treatment temperature after 4 min of irradiation (below 43 °C) due to the poor solubility and unstable property of IR780. The M$_{(I+C)}$@CaP$_{(p)}$ NP group exhibited an enhanced photothermal conversion effect, and the tumor temperature increased to 42 °C with only 20 s of irradiation. Surprisingly, owing to the RGD-mediated enhanced tumor cell uptake, the $^RM_{(I+C)}$@CaP$_{(p)}$ group showed a dramatic temperature increase to 44.4 °C after 20 s of laser irradiation, exhibiting favorable mild photothermal activity.

Subsequently, the luciferase activity of PANC-1-luc cells within in situ carcinoma xenografts was measured to test the therapeutic efficiency of the pure drug and $^RM_{(I+C)}$@CaP nanosystems with/without partzymes in vivo (Fig. 8c). The quantitative analysis of luciferase intensity is shown in Fig. 8d. First, the results showed that the pure drug with laser group (I + C + P + Laser) could not effectively inhibit tumor growth, which might have been due to the easy degradation of the pure drug in vivo. In contrast, the $^RM_{(I+C)}$@CaP +Laser group effectively inhibited tumor growth for the first 10 days during the treatment period due to its excellent photothermal therapeutic effect and tumor targeting. However, the tumor showed a second round of growth after the 20th day, and the total flux changed from 5.09 × 10$^8$ p/s to -2.72 × 10$^9$ p/s, which might be attributable to acquired drug and heat resistance. Surprisingly, when partzyme was encapsulated into the nanosystems, the therapeutic efficacy of the $^RM_{(I+C)}$@CaP$_{(P)}$ +Laser group increased significantly and continued until Day 40, and the total flux equaled 5.04 × 10$^8$ p/s. It is possible that after the HSP70 gene was silenced by the MNAzyme, the tumor was unable to produce heat resistance, and the silencing of miRNA-21 also promoted the expression of the tumor suppressor gene PTEN protein and accelerated the apoptosis of the tumor. Furthermore, the photographs of tumors in situ (Fig. 8e) and tumor weight (Fig. 8f) indicated that the $^RM_{(I+C)}$@CaP$_{(P)}$ +Laser group had the best therapeutic effect, and its tumor weight was 0.049 g, which was ~50% of that of the PBS group. The TUNEL assay also demonstrated the therapeutic efficiency of partzyme (Fig. 8g), and the $^RM_{(I+C)}$@CaP$_{(P)}$ +Laser group revealed an 18% greater TUNEL signal than the $^RM_{(I+C)}$@CaP without partzymes +Laser group (Fig. 8h).

As mentioned previously, HSP70 can protect healthy cells from high temperature, but it is also the resistance mechanism for cancer cells against photothermal therapy (Fig. 9a). Therefore, smart nanomachines that can process biological signal input (miRNA-21) will significantly increase the selectivity and efficiency of PTT (Fig. 9b). To further verify the tumor-targeted silencing function of our nanodevices, the gene and protein expression levels of HSP70 were evaluated in both tumor and paracancerous tissues. Pure drug and $^RM_{(I+C)}$@CaP nanosystem with/without partzymes groups were used.

For tumor tissue, the HSP70 protein expression level in the pure drug group increased by ~66% under laser irradiation (Fig. 9c, d) compared with that in the PBS +Laser group. Moreover, the $^RM_{(I+C)}$@CaP (without partzymes) +Laser group also showed ~34% HSP70 upregulation. Previous in vitro experiments showed that

$^RM_{(I+C)}$@CaP could downregulate HSP70 by regulating mitochondrial function (Fig. 9e). However, these in vivo experiments suggested that the nanomachine could mediate more efficient photothermal conversion efficiency than the pure drug, which exacerbated the production of HSP70 (Fig. 9c, d). Hence, the enhancement of HSP70 expression could explain how the PTT resistance caused a second round of tumor growth (Fig. 9d). Importantly, the nanocarrier loaded with partzymes ($^RM_{(I+C)}$@CaP$_{(P)}$) could achieve effective downregulation of HSP70 even under laser irradiation (Fig. 9c, d), which proved the successful MNAzyme assembly. The results of the western blot experiment also provided ample evidence of the silencing effect of MNAzymes on HSP70 (Supplementary Fig. 20).

Nevertheless, in healthy pancreatic tissue, the HSP70 levels at the outer edge of healthy tissue were elevated in the pure drug and $^RM_{(I+C)}$@CaP groups as well as in the $^RM_{(I+C)}$@CaP$_{(P)}$ +Laser group. Hence, as we expected, NPs and drugs inevitably penetrated healthy paracancerous tissues, but our intelligent nanomachine was able to distinguish cancer cells from healthy cells by virtue of its tumor signal recognition function. Later, the qPCR results correlated with the above phenomenon. Compared with those in the $^RM_{(I+C)}$@CaP group, the HSP70 mRNA levels within the $^RM_{(I+C)}$@CaP$_{(P)}$ group were reduced by nearly 90% (Fig. 9e) in tumor tissue. However, for healthy tissues around tumors, our intelligent nanosystem did not affect HSP70-based self-protection, thereby remarkably improving the targeting of the treatment.

Subsequently, the MNAzyme system-mediated miR-21 silencing effect was further studied. The miR-21 content was measured through fluorescence in situ hybridization (FISH) (Fig. 9f), and the downstream protein PTEN was characterized through IHC (Fig. 9g). First, the results clearly showed that the content of miR-21 had no relationship with laser irradiation and that the control group contained the highest amount of miR-21. The content of miR-21 decreased in both the pure drug and $^RM_{(I+C)}$@CaP groups since Cur can reduce the level of miR-21. However, we found that miR-21 was almost completely silenced within the partzyme-containing $^RM_{(I+C)}$@CaP$_{(P)}$ group. These results indicated that partzymes can utilize and consume miRNA at tumor sites, which can further achieve miR-21 silencing-based gene therapy (Fig. 9g).

PTEN has been found to be a tumor suppressor with growth and survival regulatory functions, and silencing of miR-21 can mediate PTEN upregulation, which in turn inhibits tumor growth[65]. The results in Fig. 9g revealed that, for the $^RM_{(I+C)}$@CaP and I + C + P groups, the PTEN levels were all increased, which was ascribed to Cur. The $^RM_{(I+C)}$@CaP$_{(P)}$ group exhibited the highest PTEN level due to the excellent miR-21 silencing effects of both Cur and the MNAzyme system. These results reflect the cleverness of our designed MNAzyme. It can not only recognize miR-21 and use it as the building block for MNAzyme self-assembly but also silence miR-21 to achieve gene therapy.

## Discussion

For pancreatic cancer treatment, in addition to the regulation of relevant oncogenes, regulation of the immune response is also a crucial aspect that cannot be overlooked[66]. Furthermore, the high degree of fibrosis is another reason for the lack of effective drug interventions in clinical cancer treatment[67]. Therefore, in future experiments, utilizing immunocompetent animal models and observing whether our designed MNAzyme system can activate the immune response, attenuate fibrosis in pancreatic cancer, and consequently enhance the accessibility of tumor cells to nanoparticle drugs will be highly meaningful.

In this work, we successfully constructed an intelligent nanomachine for multimodule synergistic photothermal therapy with the ability to distinguish between tumor and paracancerous tissues. In vivo experiments confirmed that the nanomachine, upon processing biological signals (miRNA-21), was activated and specifically modulated HSP70 mRNA only in tumor cells while maintaining the HSP protective

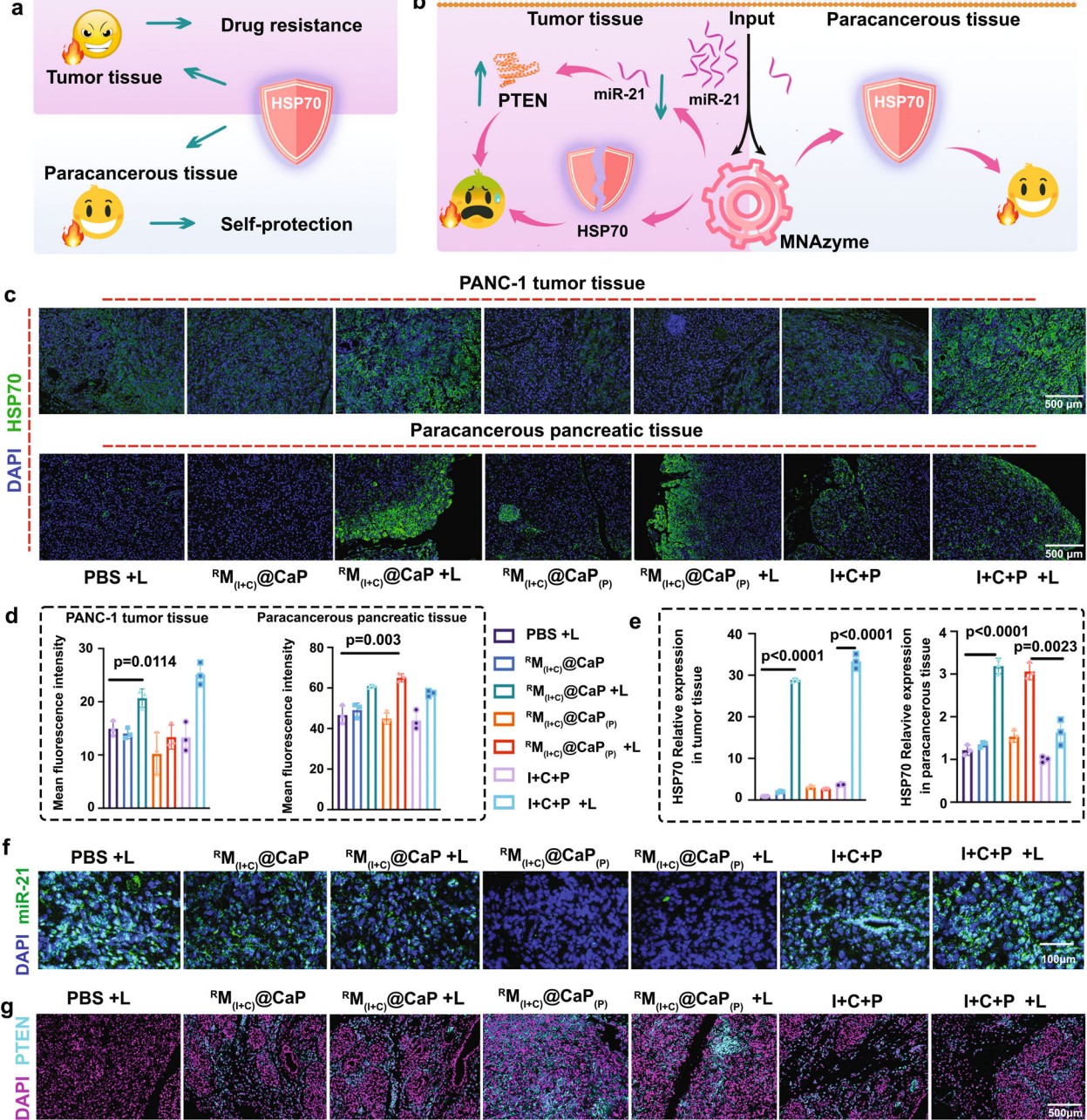

**Fig. 9 | Tumor-targeting function of the MNAzyme. a** Roles of HSPs in different cells. **b** Tumor-targeting mechanism for our intelligent nanodevice. **c** Immunohistochemical characterization of HSP70 in tumor tissues and healthy tissues. **d** HSP70 quantitated by ImageJ in tumor tissue and paracancerous tissue ($n$ = 3 independent experiments, and the data are presented as the mean values ± SDs). **e** qPCR results for HSP70 mRNA levels ($n$ = 3 independent experiments, and the data are presented as the mean values ± SDs). **f**, **g** miRNA-21 content and PTEN protein expression levels ($n$ = 3 independent experiments). All statistics were calculated using two-tailed paired $t$ tests, and all experiments (**c**, **f**, **g**) were repeated three times independently with similar results. The source data from (**d**, **e**) are provided as a Source Data file.

function in healthy cells. Furthermore, the system broke through the stereotype of MNAzyme design and simultaneously utilized the steric blocking mechanism of the miRNA/partzyme complex[36,37] as well as multiturnover catalytic rates to achieve up to 90% target gene silencing. Meanwhile, the nanocarrier was constructed with $Ca^{2+}$, which solved the problem of insufficient endogenous metal cofactors. It was also found that $Ca^{2+}$ and Cur could mediate calcium overload, mediating mitochondrial fragmentation and Cyto-C release as well as apoptosis promotion. In addition, the addition of zinc ions prevented the disadvantage of easy phase transition of calcium phosphate during storage in water and improved the stability of the formulation.

Therefore, this multisynergistic nanomachine is expected to promote the application of MNAzymes in the clinical treatment of cancer.

## Methods

### Ethical statement

Our research complied with all relevant ethical regulations. All animal protocols were performed in line with the Guidelines for the Care and Use of Laboratory Animals and approved by the Institutional Animal Care and Use Committee (IACUC) of Zhejiang Center of Laboratory Animals (ZJCLA) (Approval No. ZJCLA-IACUC-20020090). The animal experiments also follow the PREPARE Guidelines Checklist[68].

## Materials

All oligonucleotides were synthesized and HPLC-purified by Sangon Biotechnology Co., Ltd. (Shanghai, China) and are listed in Table S6. Ammonium persulfate (APS), N,N,N,N-tetramethylethylenediamine (TEMED), zinc nitrate, calcium chloride, and sodium dihydrogen phosphate were all purchased from Sigma (Finland). Tris-HCl buffer solution (pH 8.8 and 6.8), 30% acrylamide/bis, and TBS/Gly/SDS buffer were purchased from Bio-Rad Laboratories, Inc. (Finland). Solutions involved in microRNA-related experiments were prepared with DEPC-treated water (RNase-free) (Sigma, Switzerland). A BCA kit, RIPA, Tris-Triton, protease inhibitor, EDTA, and GelRed were purchased from Thermo Fisher, Finland. DSPE-PEG$_{2K}$-RGD, DSPE-PEG-COOH, and RGD were acquired from Xi'an Ruixi Biological Technology Co. Ltd. IR780 and curcumin were purchased from Macklin Inc. (China). Anti-mouse PTEN antibody, (Catalog no. ab267787), Anti-Rabbit GAPDH, (Catalog no. ab210113), Anti-mouse Anti-Cytochrome C, (Catalog no. ab13575), Anti-mouse Anti-beta Actin antibody [mAbcam 8226], Anti-rabbit Anti-pro Caspase-3, (Catalog no. ab32150), Anti-rabbit Anti-Cleaved Caspase-3 (Catalog no. ab32042) were provided from Abcam. Anti-mouse HSP70 Antibody, (Catalog no. #4872) is purchased from Cell Signaling Technology. PMCA ATPase Monoclonal Antibody (Catalog # MA3-914) was provided by ThermoFisher.

## Twenty percent polyacrylamide gel electrophoresis (PAGE) tests

Twenty percent gel solution was transferred into a plate with a shim, and the comb was inserted immediately to ensure that there were no air bubbles in the gel. The gel was allowed to set at room temperature for ~30–60 min. After coagulation, the samples were loaded into a vertical electrophoresis tank, and a prerun of ~10 min was first performed at a voltage of 5 V. Oligonucleotide samples were prepared by mixing 2 μl of sample with 8 μl of water and 2 μl of loading buffer. At least 10 μg of oligonucleotide sample was needed for loading. The sample was carefully loaded into the well without introducing any air bubbles. The gels were run at 90 V for ~2 h until the dye front was near the bottom of the gel. The glass plates were opened with a spatula to separate the upper glass plate. GelRed was diluted to 1:10,000, and the gel was stained for 5–10 min. The gel was then immersed in distilled water for 10 min to remove excess stain and reduce background staining.

## Fabrication of CaP nanoparticles

Pure CaP NPs were synthesized through a simple one-step method at room temperature by combining two solutions for 5 min. Solution A was composed of 0.42 ml of CaCl$_2$ (2.0 M) and 0.84 ml of Tris buffer (pH = 10, 10 mM) in 2 ml of Milli-Q H$_2$O. Solution B contained 0.43 ml HEPES buffer (280 mM NaCl, 15 mM Na$_2$HPO$_4$ and 50 mM HEPES) and 0.3 ml of Milli-Q H$_2$O. Solution B was added dropwise into solution A with stirring for 5 min at room temperature. Then, the NPs were washed with pure ethanol 3 times to remove water molecules. The nanoparticle stock solution was kept in ethanol.

## Observation of cell morphology with unstable CaP

PANC-1 cells were seeded in 6-well plates at a density of $1 \times 10^6$ cells in one well and cultured for 12 h. Then, the medium was replaced with 2 ml of DMEM containing Pure CaP NPs at a concentration of 50 μg ml$^{-1}$ for 12 h. After washing with PBS twice, the cells were treated with trypsin and then centrifuged at $150 \times g$ to collect the precipitate. The obtained precipitate was fixed, dehydrated, embedded, sliced, and measured by transmission electron microscopy.

## Fabrication of Zn-substituted CaP nanoparticles

*Zn-stabilized* CaP NPs were synthesized based on the abovementioned simple method at room temperature by combining two solutions for 30 min. The Zn/Ca molecular ratio was 1:20. Solution A was composed of 0.42 ml of CaCl$_2$ (2.0 M), 0.021 ml ZnCl$_2$ (2.0 M) and 0.84 ml of Tris buffer (pH = 7, 10 mM) in 2 mL of Milli-Q H$_2$O. Solution B contained 0.43 ml HEPES buffer (280 mM NaCl, 15 mM Na$_2$HPO$_4$ and 50 mM HEPES) and 0.3 ml of Milli-Q H$_2$O. Solution B was added dropwise into solution A with stirring for 30 min at room temperature.

## Fabrication of $^R$M$_{(I+C)}$ and $^R$M$_{(I+C)}$@CaP$_{(p)}$

For both drug coloadings, 50 mg of DSPE-PEG-RGD polymers were dissolved in 5 ml of organic solvent composed of 70% ethanol and 30% chloroform. Different concentrations of drugs with different IR780/Cur ratios (0:2; 0.5:2; 1:2; 2:2; 4:2; 8:2 mg) were also dissolved in the abovementioned organic solvent. The solution was then placed into a rotary evaporator for film formation. Subsequently, 10 ml of water was added for hydration for 24 h and centrifuged for 5 min at 1400 g to remove the unencapsulated drugs[69] since both drugs possessed very low water dissolving ability. Then, 10 ml $^R$M$_{(I+C)}$ micelles were prepared. To study the drug-loading capacity of IR780 and Cur separately, we explored the loading amount of 50 mg DSPE-PEG polymer with gradient drug concentrations. The specific method was the same as above, with concentration ranges of 2 to 10 mg for IR780 and 0.5 to 3 mg for Cur. Drug-loading capacity = (weight of drug input minus the undissolved drug/weight of nanoparticles) × 100%, drug loading efficiency = (weight of drug in nanoparticles/weight of drug used for nanoparticle preparation) × 100%. For $^R$M$_{(I+C)}$@CaP$_{(p)}$ NPs, a 1 ml micelle solution with 1 ml of Milli-Q H$_2$O was used to replace the 2 ml of Milli-Q H$_2$O in the abovementioned solution A when preparing the CaP NPs. Before CaP mineralization, partzymes at different concentrations (1, 10, 50, 100, and 200 μM) were precultured with solution A. The M$_{(I+C)}$ and M$_{(I+C)}$@CaP$_{(p)}$ NPs were fabricated through the same procedure, and DSPE-PEG-RGD was replaced by DSPE-PEG. The partzymes were modified with Cy3, enabling the measurement of nonencapsulated DNAzymes in the supernatant after centrifugation using UV spectrophotometry. The drug loading for IR780, Cur and DNAzymes was carried out through UV–Visible spectroscopy at 780, 425 and 560 nm, respectively. For CaP-containing NPs, we used centrifugation to isolate the NPs from the reaction mixture and subsequently washed the NP solution multiple times with a suitable solvent to remove any impurities.

## Characterization of the nanoparticles

The particle sizes of CaP, $^R$M$_{(I+C)}$, and $^R$M$_{(I+C)}$@CaP$_{(p)}$ NPs were determined using dynamic light scattering with a Zetasizer Nano ZS. For each measurement, the sample (1.0 ml) was placed in a disposable polystyrene cuvette. The nanocarrier surface ζ-potential was measured with a Zetasizer Nano ZS by using disposable cells. Both the size and ζ-potential were recorded as the average of three measurements. The TEM samples were prepared by dropping the particle solution on top of carbon-coated copper grids and then drying it in air prior to imaging. Fourier transform infrared spectroscopy (FTIR) was performed in the 4000–1000 cm$^{-1}$ range with a resolution of 4 cm$^{-1}$ at room temperature by using a Thermo Nicolet IS10 spectrometer.

## Photothermal properties of $^R$M$_{(I+C)}$@CaP$_{(p)}$ NPs

$^R$M$_{(I+C)}$@CaP$_{(p)}$ NPs were diluted with PBS (concentration of IR780 was from 0 to 100 μg/ml), and 1 ml of sample was added to a cuvette and irradiated with an 808 nm NIR laser source at 1.0 W/cm$^2$ for 5 min. The temperature changes of the samples were recorded by the infrared imaging device at intervals of 30 s. Temperature elevations of free IR780 (dissolved in PBS), $^R$M$_{(I+C)}$ and $^R$M$_{(I+C)}$@CaP$_{(p)}$ NPs over four NIR on–off irradiation cycles were conducted through a 1.0 W cm$^{-2}$ 808 nm laser with 50 μg/ml IR780. The photobleaching properties of free IR780, $^R$M$_{(I+C)}$ and $^R$M$_{(I+C)}$@CaP$_{(p)}$ NPs were investigated by diluting the IR780 content in each group to 50 μg/ml, and laser irradiation (808 nm, 1.0 W/cm$^2$) was continued for 5 min. Then, the UV absorbance of each group was monitored. The irradiation process was conducted 3 times.

## Drug release experiment

One milliliter of $^RM_{(I+C)}@CaP_{(p)}$ NPs (containing 0.4 mg IR780, 0.2 mg Cur and 200 µM partzymes) was tied into a dialysis bag and released under 20 ml sink conditions. The release medium contained 2% Tween-80. There were four drug release conditions: pH = 7, 37 °C; pH = 7, 45 °C; pH = 6, 37 °C and pH = 6, 45 °C. At each time point, 1 ml of release medium was extracted for drug concentration measurement and then added back to the system. The absorbance characteristic peak wavelengths of IR780, Cur and partzymes were observed using a NanoDrop™ 2000 spectrophotometer. For MNAzyme, 1 µl of released medium was mixed with 1 µl of miRNA-21 (10 µM). Then, 2 µl of 10 µM HSP70 mRNA was added to the reaction, resulting in a 4 µl volume, with a maximum of 2.5 µM MNAzyme to mediate 5 µM HSP70 mRNA cleavage.

## Cell culture and maintenance

The pancreatic cancer cell line PANC-1 and normal human dermal fibroblast NHDFs were grown in DMEM with 10% FBS at 37 °C. Cells were passaged 2–3 times a week once they reached 90–100% confluency. The human pancreatic cancer cell line PANC-1 was purchased from ATCC® (CRL-1469™), and normal human dermal fibroblast (NHDF) cells were purchased from PromoCell®. The cell lines were directly purchased without further authenticated, and the cell line were tested negative for mycoplasma contamination.

## Cellular uptake study

Cells were incubated in 6-well plates ($1 \times 10^5$ cells per well) overnight. When the cells were attached, solutions of free IR780 + Cur + partzyme, $M_{(I+C)}@CaP_{(P)}$, and $^RM_{(I+C)}@CaP_{(P)}$ were utilized to replace the cell growth media. All the groups were kept at a concentration of 4 µg/ml for IR780, 2 µg/ml for Cur and 2 µM for partzymes at 37 °C. After incubation for 1 h and 8 h, the cells were observed under CLSM. Additionally, the cells were collected with trypsin and washed with PBS. Cellular uptake was determined with a BD LSRFortessa flow cytometer (BD Biosciences), and the results were analyzed with FlowJo_V10. The gate was defined for live cells only; 10,000 cells were recorded per sample. The concentrations of $Ca^{2+}$ in PANC-1 cells were measured using inductively coupled plasma–optical emission spectrometry (ICP–OES). The same experimental groups were treated with final concentrations of IR780 at 4 µg/ml, Cur at 2 µg/ml, and partzyme at 2 µM. After washing the cells with PBS twice, the cells were treated with trypsin and collected through centrifugation. Finally, 1.0 ml of concentrated nitric acid was added, and the samples were measured by using ICP–OES to determine the $Ca^{2+}$ concentration.

## Fluorescence in situ hybridization (FISH) detection

A fluorescence in situ hybridization (FISH) kit was purchased from Guangzhou Exons Biological Technology Co., Ltd. The probe was diluted with hybridization buffer and denatured at 85 °C for 2 min. Then, the probe was used for hybridization with 4% paraformaldehyde-fixed cells for 72 h. The cells were labeled with DAPI (Amax = 358, Emax = 461). The fluorescence of the probe was Cy5.5 (Amax = 683, Emax = 703). The relative intensity of each group was calculated with ImageJ.

## Lysosome escape

IR780 and LysoTracker Green were used as the probe of the drug and lysosome, respectively. PANC-1 cells were seeded on confocal dishes overnight. Then, the lysosomal escape profile of the $^RM_{(I+C)}@CaP_{(p)}$ groups (±Laser) by PANC-1 cells at 2 h and 24 h was determined. For both groups, after NPs and cells were cultured together for 2 h and 24 h, the cell media were replaced, and a 1 W laser was applied for 5 min. After laser irradiation, the cells were incubated with LysoTracker Green for another 2 h, after which a 5 min incubation was performed with 4% paraformaldehyde. Then, the cells were observed under a fluorescence confocal microscope. The colocalization scatterplots of each group were calculated by ImageJ.

## Western blot analysis

After the cells were incubated with different groups of nanomaterials for 48 h, the cells were lysed with RIPA and Tris-HCl lysis buffer. RIPA was used for whole-cell lysis and is suitable for destroying mitochondria. For 2 ml of RIPA buffer, 20 µl of 1% protease inhibitor and 20 µl of 1% 0.5 M EDTA were added. Tris-HCl was used for cytoplasmic cleavage. The final concentration of Tris was 20 mM. Twenty microliters of 1% protease inhibitor and 20 µl of 1% 0.5 M EDTA were added to 2 ml of Tris-HCl. After 1 h of lysis, the medium buffer was centrifuged at $16,000 \times g$ for 10 min, and the supernatant was taken for quantification. Cell proteins were separated by SDS–PAGE gradient gel and transferred to PVDF membranes. After adding protein loading buffer, the samples were mixed well and fully denatured at 98 °C for 10 min. The membranes were incubated with primary antibodies against PTEN, HSP70, Cyto-C, β-Actine, Pro-Caspase 3, cleaved-caspase 3 and GAPDH (Abcam, Cambridge, UK) (1:1000) at 4 °C overnight. Then, the membranes were incubated with the secondary antibody at 37 °C for 1 h. A GelDoc Go Gel Imaging System (Bio-Rad) was used to evaluate protein bands. ImageJ software was used for fluorescence quantitative analysis.

## Colocalization between IR780 and mitochondria

Eight groups, including the PBS ±Laser, I + C ±Laser, $M_{(I+C)}@CaP_{(p)}$ (without RGD) ±Laser and $^RM_{(I+C)}@CaP_{(p)}$ (with RGD) ±Laser groups, were established and marked from 1 to 8. The laser groups were under $1\,W\,cm^{-2}$ 808 nm irradiation for 5 min. IR780 exhibited red fluorescence, and mitochondria were labeled with MitoTracker Green. The mitochondrial tracker was purchased from Beyotime Biotechnology Co., Ltd. After nanosystems with a concentration of IR780: 4 µg/ml, Cur: 2 µg/ml, and each partzyme: 2 µM were cultured with PANC-1 cells for 6 h, the medium was replaced, and a laser was applied for 5 min. The cells were then cultured with the mitochondrial tracker for another 2 h. Then, the samples were observed under fluorescence confocal microscopy. ImageJ software was used for fluorescence quantitative analysis.

## ROS detection

PANC-1 cells were cultured in confocal dishes at $1 \times 10^5$ cells per well. When the PANC-1 cells reached 70–80% confluency, they were treated under different conditions, including incubation with PBS ±Laser, I + C ±Laser, $M_{(I+C)}@CaP_{(p)}$ (without RGD) ±Laser and $^RM_{(I+C)}@CaP_{(p)}$ (with RGD) ±Laser for 6 h at 37 °C. After 6 h of incubation, dichlorodihydrofluorescein diacetate (DCFH-DA) solution was added to the cell medium[70]. The ROS signals were quantitatively analyzed with ImageJ software, which indicated the intracellular $1O_2$ levels generated under different experimental conditions.

## $Ca^{2+}$ concentration visualization

PANC-1 cells were cultured in confocal dishes at $1 \times 10^5$ cells per well. When PANC-1 cells reached 70–80% confluency, they were treated under different conditions, including incubation with PBS ±Laser, I + C ±Laser, $M_{(I+C)}@CaP_{(p)}$ (without RGD) ±Laser and $^RM_{(I+C)}@CaP_{(p)}$ (with RGD) ±Laser for 6 h at 37 °C. After 6 h of incubation, the Fluo-3 AM solution was subsequently added to the cell medium. The concentration of the Fluo-3 AM probe in all samples was 1.0 µM. ImageJ software was used for fluorescence quantitative analysis.

## ΔΨm detection by tracking of JC-1 monomers and aggregates

To measure the changes in mitochondrial membrane potential, PANC-1 cells were seeded in a 6-well plate at a density of $2*10^5$ cells per well in 2.0 ml of FPS-DMEM and cultured for 24 h. The medium was then

replaced with 1.0 ml of PBS ±Laser, I + C ±Laser, $M_{(I+C)}@CaP_{(p)}$ (without RGD) ±Laser and $^{R}M_{(I+C)}@CaP_{(p)}$ (with RGD) ±Laser at the same concentrations, IR780: 4 μg/ml, Cur: 2 μg/ml, each partzyme: 2 μM, and cultured for 12 h. After washing with PBS twice, the cells were stained with JC-1 and measured by CLSM. In this detection, 1.0 ml of JC-1 working solution was added to cells containing 1.0 ml of DMEM.

## Observation of mitochondrial morphology

To measure mitochondrial morphology, PANC-1 cells were seeded in a 100 mm × 20 mm dish at a density of $3.0 \times 10^{6}$ cells per dish and cultured for 24 h. The medium was then replaced with 5.0 ml of PBS ±Laser, I + C ±Laser, $M_{(I+C)}@CaP_{(p)}$ (without RGD) ±Laser and $^{R}M_{(I+C)}@CaP_{(p)}$ (with RGD) ±Laser at the same concentrations, IR780: 4 μg/ml, Cur: 2 μg/mL, each partzyme: 2 μM, and cultured for 12 h. After being washed with PBS twice, the cells were treated with trypsin and then centrifuged at $150 \times g$ to collect the precipitate. The cells were then fixed for 1 day at room temperature. After fixation, the cells were washed in the same buffer and postfixed in 1% osmium tetroxide in 0.1 M sodium cacodylate buffer (pH 7.4) for 1 h at room temperature. The cells were then washed again with the same buffer and dehydrated in a graded series of ethanol solutions. The obtained cells were then embedded in epoxy resin (Epon 812) and polymerized for 48 h at 60 °C. The polymerized blocks were then sectioned into ultrathin sections using a Leica EM UC7 ultramicrotome. The sections were collected on copper grids and stained with uranyl acetate and lead citrate before being examined under a JEM-1400 Plus transmission electron microscope (TEM). During TEM imaging, at least 3 random images were taken at a magnification of 5000–10,000× to observe the mitochondrial ultrastructure. The mitochondrial morphology was analyzed based on the presence or absence of cristae, the shape of the mitochondrial membrane, and the size of the mitochondria. The sample preparation was carried out in the Laboratory of Electron Microscopy, Medisiina C, 2nd floor, University of Turku. A JEM-1400 Plus transmission electron microscope was used to monitor mitochondrial morphology.

## Annexin V-APC/PI apoptosis detection

Cells were digested with trypsin solution and aspirated until the adherent cells could be blown down by gentle pipetting. Then, the cells were transferred into a centrifuge tube and centrifuged at $1000 \times g$ for 5 min, and the supernatant was discarded. Subsequently, the cells were gently resuspended in PBS and counted. A total of 50,000–100,000 resuspended cells were collected and centrifuged at $1000 \times g$ for 5 min. Then, 195 μl of Annexin V-APC binding solution was added to gently resuspend the cells. Then, 10 μl of propidium iodide staining solution containing Annexin V-APC was added and mixed gently. After incubation for 10–20 min at room temperature (20–25 °C) in the dark, the cells were placed in an ice bath. Apoptosis was determined with a BD LSRFortessa flow cytometer (BD Biosciences), and the results were analyzed with FlowJo V10; 10000 cells were recorded per sample.

## Cytotoxicity assay

The concentrations of the therapeutics were as follows: IR780: 4 μg/ml, Cur: 2 μg/ml and partzymes: 2 μM. The stock solution of $^{R}M_{(I+C)}@CaP_{(p)}$ NPs (containing 0.4 mg IR780, 0.2 mg Cur and 200 μM partzyme per ml) was diluted 100-fold to obtain 4 μg/ml IR780 and 2 μg/ml Cur, and there was 2 μM partzyme. After 24 h of incubation, the cells were subjected to standard WST-1 experiments. All dilutions for the cell viability assay were prepared in cell growth medium. After incubation with free drug or nanoparticles, 10 μl of WST-1 reagent was added to each well, and the cells were incubated for 2 h at 37 °C with 5% $CO_2$. After incubation, the absorbance was measured with a Varioskan Flash Multimode Reader (Thermo Scientific Inc., Waltham, MA, USA) at 440 nm.

## Detection of IR780 by HPLC

The detection wavelength of IR780 was 700 nm with a Waters 2695 Alliance HPLC. The mobile phase flow rate was set at 1.0 ml/min. The chromatographic column used was an Eclipse plus C18 column. Mobile phase A contained methanol and 0.2% FA, and mobile phase B contained water and 0.2% FA. The ratio was 4:1 (A:B, v/v).

## Orthotopic model construction procedure

The experimental procedure was based on previously reported methods[71] with certain improvements. BALBc-nu (6–8 weeks old) nude mice were anesthetized and positioned in a left lateral recumbent position. Following routine skin disinfection, a 1 cm oblique incision was made just below the right costal margin, allowing access to the abdominal cavity. Afterward, the abdominal cavity was opened, revealing the spleen and pancreas. A prepared cell suspension was then precisely injected into the pancreas, with attention to minimizing tissue damage. Hemostasis was performed if necessary, and the incision was sutured to close the abdominal cavity. Finally, the surgical site was disinfected once more, and the mouse was placed in a warm incubator or recovery area for postoperative care, all while adhering to ethical and animal care guidelines. The mice feeding conditions of mice are: 12/12 day and night alternating, humidity at 50–60%, temperature at 22–26 °C.

## In vivo biodistribution

The I + C + P, $M_{(I+C)}@CaP_{(p)}$ and $^{R}M_{(I+C)}@CaP_{(p)}$ groups were used to observe the biodistribution of nanodevices after intravenous administration. The concentrations of the therapeutics were as follows: IR780: 2 mg/kg, Cur: 1 mg/kg and partzymes: 1 μmol/kg. In vivo imaging was performed at 1, 8, 24 and 48 h timepoints. On the 48th day, the mice from each group were sacrificed, and the tumor, heart, liver, spleen, lungs and kidneys were collected for in vitro fluorescence detection.

## In vivo IR thermography

Four groups, including the PBS, I + C + P, $M_{(I+C)}@CaP_{(p)}$ and $^{R}M_{(I+C)}@CaP_{(p)}$ groups, were established. First, the nude mice were anesthetized, and the tumor site was observed and marked with an in vivo imaging system. Then, the mice were fixed, and at 12 h after administration, the tumor site was irradiated with an 808 nm laser.

## In vivo anticancer efficacy analysis

PANC-1 cells containing luciferase were injected into the pancreases of BALB/c-nu (6–8 weeks old) female mice. The mice were divided into seven groups: the PBS + Laser, $^{R}M_{(I+C)}@CaP$, $^{R}M_{(I+C)}@CaP$ + Laser, $^{R}M_{(I+C)}@CaP_{(p)}$, $^{R}M_{(I+C)}@CaP_{(p)}$ + Laser, I + C + P, and I + C + P + Laser groups. The concentrations of the therapeutics were as follows: IR780: 2 mg/kg, Cur: 1 mg/kg and partzymes: 1 μmol/kg. For the laser treatment groups, the temperature of the tumor site was monitored in real time by the in vivo imaging system. Radiation was stopped when the temperature reached ~45 °C and continued when the temperature dropped. The total treatment time was 5 min. The tumor volumes were calculated using the in vivo imaging system software by monitoring the total flux of bioluminescence. After 40 days of treatment, the tumors connected to the spleen were taken for comparison. For orthotopic tumor models, the maximum tumor diameter allowed was 1.5 cm. Humane endpoints encompass tumor burden exceeding 10% of normal body weight, animal weight loss surpassing 20% of the normal animal weight, and persistent self-harm by the animal. Humane endpoints involve euthanasia through cervical dislocation under deep anesthesia. All animal procedures adhere to the guidelines approved by the Institutional Animal Care and Use Committee (IACUC) of Zhejiang Center of Laboratory Animals (ZJCLA). Pre-experimentation ensures that orthotopic tumor weights align with established

standards during the treatment period. Also, the ex vivo photographs in this experiment confirmed that none of the tumors exceeded this limit.

### TUNEL, HSP70, and PTEN staining

After the slices were dried slightly, a circle around the tissue was drawn with a histochemical pen (to prevent liquid from flowing away), proteinase K was added to the circle to cover the tissue, and the tissue was incubated in a 37 °C incubator for 25 min. The slides were placed in PBS (pH 7.4) on a decolorizing shaker and washed 3 times with 5 min shaking each time. After the slices were slightly dried, working rupture solution was added to the circle to cover the tissue, and the slides were incubated at room temperature for 20 min. The slides were placed in PBS (pH 7.4) and washed 3 times on a decolorizing shaker. Subsequently, for TUNEL staining, appropriate amounts of reagent 1 (TdT) and reagent 2 (dUTP) were taken from the TUNEL kit, mixed 1:9 and added to the circle to cover the tissue. Then, the slices were flattened in a wet box and incubated for 2 h at a constant temperature of 37 °C. A small amount of water was added to the humid box to maintain humidity. For HSP70 staining, HSP70 antibody was added to the sample after dilution at 1:100–1:500. After washing with PBS three times, the secondary antibody, Cy3-conjugated goat anti-rabbit IgG, was added, and the slides were incubated for 60 min. For PTEN staining, the tumor sections were stained with anti-PTEN antibody. Later, the sections were all washed with PBS (pH = 7.4) 3 times for 5 min each. After removing the PBS, the secondary antibody, Cy3-conjugated goat anti-rabbit IgG, was added, and the sections were incubated for 60 min. DAPI staining solution was also added to the circle, and the sections were incubated for 10 min at room temperature in the dark. Then, the slides were placed in PBS (pH 7.4) and washed 3 times. Finally, the slices were observed under a fluorescence microscope. The positivity rate of the experiment was calculated with ImageJ.

### Analysis of HSP70 levels by qPCR

The intratumoral and paracancerous tissue HSP70 mRNA levels were assessed with qPCR. The forward primer and reverse primer for HSP70 were 5′-ACCAAGCAGACGCAGATCTTC-3′ and 5′-CGCCCTCGTA-CACCTGGAT-3′, respectively. The forward primer and reverse primer for GAPDH were 5′-GGCATGGACTGTGGTCATGA-3′ and 5′-GGCATG-GACTGTGGTCATGA-3′, respectively.

### Statistics and reproducibility

All results are expressed as the mean value ± standard deviation. Student's unpaired $t$ test (two-tailed) was employed for comparisons between different groups. All statistical analyses were performed using GraphPad Prism 9.0. and $p < 0.05$ was considered to indicate statistical significance. The measurements were taken from distinct samples. No statistical method was used to predetermine the sample size. No data were excluded from the analyses. Data collection and analysis were not performed blind to the conditions of the experiments. All results may be duplicated from the available source data files.

### Reporting summary

Further information on research design is available in the Nature Portfolio Reporting Summary linked to this article.

## Data availability

The authors declare that all the data supporting the findings of this study are available within the Article, Supplementary Information or Source Data file. The source data generated in this study have been deposited in the Figshare database at: https://figshare.com/articles/dataset/Source_Data_xlsx/24271477. Source data are provided with this paper.

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

## Acknowledgements

This work was supported by the National Science Foundation (Grant No. 82372145 (H.Z.)). It was also supported by the Research Fellow (Grant No. 353146 (H.Z.)), Project (Grant No. 347897 (H.Z.)), Solution for Health Profile (Grant No. 336355 (H.Z.)), and InFLAMES Flagship (Grant No. 337531 (H.Z.)) grants from the Academy of Finland, Finland, The China Food and Health International Pilot Project (H.Z.) funded by the Finnish Ministry of Education and Culture, The Leading Talents in Scientific and Technological Innovation from Zhejiang Provincial Ten Thousand Talents Plan (Grant No. 2019R52021 (X.S.)), The Key Research and Development Program of Zhejiang Province (Grant No. 2021C03120 (X.S.)). The Key Research and Development Program of Wenzhou (Grant No: ZY2021003 (X.S.)), the National Natural Science Foundation (Grant Nos. 82272172 (W.S.), 81972261 (W.S.), the Medical Health Science and Technology key Project of Zhejiang Provincial and Ministry Health Commission (Grant No. WKJ-ZJ-2322 (W.S.)), the National Natural Science Foundation of China (Grant No. 82071945 (S.Z.)), Shanghai Committee of Science and Technology, China (Grant No. 21S31905400 (S.Z.)), and the Shanghai Anticancer Association EYAS PROJECT (Grant No. SACA-CY22C07 (D.Z.)). M.R. (CSC202207960005) and J.Y. (CSC202107960001) were sponsored by the China Scholarship Council. We are grateful to Qian Zhou, Ph.D., from the Fudan University Shanghai Cancer Center. We are grateful to Gösta Branders research fund, Åbo Akademi Research Foundation (Gösta Branders forskningsfond, Stiftelsen för Åbo Akademi).

## Author contributions

J.Y. and H.Z. conceived the idea and wrote the manuscript. J.Y., X.M., D.L., and M.R. were responsible for carried out the experiments, material synthesis and animal experiments. D.Z. and S.Z. assisted in manuscript revision related to pancreatic cancer. X.C. helped to conduct parts of the characterization. W.S., X.S., and H.Z. supervised the project and helped with language editing. All authors discussed the results and have given approval to the final version of the paper.

## Competing interests

The authors declare no competing interests.

## Additional information

[1]Joint Centre of Translational Medicine, Wenzhou Key Laboratory of Interdiscipline and Translational Medicine, The First Affiliated Hospital of Wenzhou Medical University, Wenzhou, China. [2]Department of General Surgery, The First Affiliated Hospital of Wenzhou Medical University, Wenzhou, Zhejiang, China. [3]Pharmaceutical Sciences Laboratory, Faculty of Science and Engineering, Åbo Akademi University, Turku, Finland. [4]Turku Bioscience Centre, University of Turku and Åbo Akademi University, Turku, Finland. [5]Department of Gastrointestinal Surgery, The Second Affiliated Hospital and Yuying Children's Hospital of Wenzhou Medical University, Wenzhou 325027 Zhejiang, China. [6]Department of Ultrasound, Fudan University Shanghai Cancer Center, Shanghai 200032, PR China. [7]These authors contributed equally: Jiaqi Yan, Xiaodong Ma, Danna Liang, Meixin Ran. ✉e-mail: fame198288@126.com; shenxian@wmu.edu.cn; hongbo.zhang@abo.fi

