## [Peer Review File · Nature Communications]

Reviewers' Comments:

Reviewer #2:

Remarks to the Author:

This research manuscript reporting: An autocatalytic deoxyribozyme based nanomachine for orthotopic pancreatic tumor targeted and multilevel synergistic photothermal therapy. This is an interesting paper which is supported with various experimental data. There are a few queries from the reviewer:

1) Statement in the manuscript: Approximately 35% IR780 and 43% Cur were released at pH=6, 45 °C, whereas only 9% IR780 and 14% Cur were released at pH=6, 37 °C. A similar phenomenon was also found in the pH=7 group. More than 20% IR780 and 30% of curcumin were released at pH=7 45°C. However, less than 5% of IR780 and 10% of curcumin were released at pH=7 at 37°C. These results reflected that the drug was loaded inside micelles, and its release was mainly dependent on the temperature. The overall low drug releasing profile for IR780 and Cur (less than 50%) was attributed to the lack of solubility in the release media.

Question: The authors should explain why temperature affects the curcumin release. Did the authors use thermal-responsive carriers to encapsulate curcumin? In addition, the authors should clarify further on how drug release in different temperature is evidence of drug loaded inside micelles.

2) Statement in the manuscript: Meantime, Zn²⁺-substitute CaP(p) showed a strong positive charge (10.01 mV) due to the uncoordinated calcium and zinc ions on its surface.

Question: Authors are requested to provide further clarification and data to support the presence of Zn on the biomaterials surface

3) In Table S5 reports stability studies for biomaterials (Nano systems) at refrigerated and accelerated-temperature conditions.

Question: The zeta potential values are relatively closed to zero mV and are indicative of unstable nanoparticles. The authors should probe further on other factors that influence stability of nanoparticles. In addition, authors should perform statistical analysis for all the reported data (Table S5). Compare the data obtained at different storage temperature.

Reviewer #3:

Remarks to the Author:

The authors have replied the comments that the reviewers addressed. It can be accepted for publication.

Reviewer #4:

Remarks to the Author:

This manuscript was responsive to the prior critiques from reviewers and seems to have addressed all comments. With regard to the in vivo models of pancreatic cancer used, the authors employ orthotopic implantation of human PDAC cell lines into the pancreas. These cells express luciferase, enabling effective in vivo imaging of tumors to verify their presence and impact of intervention.

This approach has advantages given that the tumors are located in the natural organ, which is superior to subcutaneous models. The data presented are also transparent, showing each individual animal as a representative image and the error bars on graphs to capture variability.

Some minor additions are recommended to improve the discussion/data on these in vivo models:

1. Methods, more details related to the implantation of the cells into the pancreas is needed, or at least a reference to the paper from which they derived this method. This is usually done via a surgical procedure and this detail is not evident in the paper

2. Discussion, the authors should acknowledge the limitation of using immunodeficient mice to implant human tumors. Although necessary it does not likely recapitulate the immune features and desmoplasia of human tumors that are really important in this disease.

Respond letter for reviewer

Response to All Reviewers

We express our gratitude to the reviewers for their valuable and perceptive feedback. We have meticulously addressed each comment in this comprehensive response. **The reviewers' comments are presented in black text, while our responses are provided in blue.** Revised sections of the main text and Supporting Information are indicated by yellow highlights, and modifications to the methods section have been incorporated to align with the updated figure.

Reviewer 2

This research manuscript reporting: An autocatalytic deoxyribozyme based nanomachine for orthotopic pancreatic tumor targeted and multilevel synergistic photothermal therapy. This is an interesting paper which is supported with various experimental data. There are a few queries from the reviewer:

We appreciate the positive evaluation of our manuscript by the reviewers. We also extend our gratitude for the insightful questions raised by the reviewers, as these inquiries have significantly contributed to enhancing the quality of our article.

1) Statement in the manuscript: Approximately 35% IR780 and 43% Cur were released at pH=6, 45 °C, whereas only 9% IR780 and 14% Cur were released at pH=6, 37 °C. A similar phenomenon was also found in the pH=7 group. More than 20% IR780 and 30% of curcumin were released at pH=7 45 °C. However, less than 5% of IR780 and 10% of curcumin were released at pH=7 at 37 °C. These results reflected that the drug was loaded inside micelles, and its release was mainly dependent on the temperature. The overall low drug releasing profile for IR780 and Cur (less than 50%) was attributed to the lack of solubility in the release media.

Question: The authors should explain why temperature affects the curcumin release. Did the authors use thermal-responsive carriers to encapsulate curcumin?

We thank the reviewer for raising an important point regarding the enhanced drug release from ${}^R\text{M}_{(I+C)}@CaP_{(p)}$ at high temperatures.

In this study, we explored the heat-promoted situation, not due to inherent thermo-responsive nature of the drug-carrier, but due to the heat generated by IR780 upon laser irradiation, will cause localized temperature elevation. The higher temperature may cause enhanced drug release. Hence, 45°C was used to simulate this condition.

For better understanding the release of the drug from DSPE-PEG₂₀₀₀ micelles at different temperatures, we conducted several experiments and literature reviews. There should be two mechanisms for curcumin release from ${}^R\text{M}_{(I+C)}@CaP_{(p)}$ NPs. The first mechanism is the diffusion of the drug from the hydrophobic core of the micelle. The second mechanism is the

dissociation of the micelle or NPs, leading to drug release. From the experimental observations (Figure 1), we found that the size and morphology of ${}^R\text{M}_{(I+C)}@CaP_{(p)}$ NPs remained unchanged at both 37°C and 45°C (pH = 7) after 48h incubation, indicating that the ${}^R\text{M}_{(I+C)}$ micelles and ${}^R\text{M}_{(I+C)}@CaP_{(p)}$ NPs did not dissociate. Therefore, the increased curcumin release from pure DSPE-PEG₂₀₀₀ micelles could be attributed to the increased thermal kinetic energy of molecules diffusion.

Figure 1. Morphology of NPs at different temperatures

On the other hand, regarding DSPE-PEG₂₀₀₀ micelles, literature review reveals that DSPE-PEG₂₀₀₀ micelles are highly stable and exhibit a low critical micelle concentration of approximately $1 \times 10^{-6} \text{ M}^{-1}$. However, elevating the temperature can cause a slight increase in the size of the micelle (Figure 2), which in turn may lead to a decrease in the micelle's density. This size changing effect is limited in our case since ${}^R\text{M}_{(I+C)}@CaP_{(p)}$ contained CaP shell. However, this reduced density could potentially enhance the leakage ability of the drug from the ${}^R\text{M}_{(I+C)}$ structure.

Figure 2. Hydrodynamic diameter of DSPE-PEG(2000) micelles is shown as a function of temperature. Micelle size remains consistent with small, spheroidal aggregates both above and below their transition temperature ¹.

Furthermore, DSPE-PEG₂₀₀₀ micelles maintain a fluid core at room temperature ². DSC studies indicate that a transition involving lipid chain melting in the core of DSPE-PEG₂₀₀₀ micelles occurs around 12°C ¹ (Figure 3). And higher temperatures do not induce further phase transitions in this micelle.

Figure 3. An endothermic transition upon heating is seen in DSPE-PEG(2000) micelles at 12.8° ¹.

Consequently, at 37°C and 45°C, the micellar structure should have no phase transitions. The accelerated drug release at 45°C likely attributed to increased thermal energy enhancing molecular motion, thereby facilitating drug release. However, the results indicated that after 48 hours of release at 45°C, the system tends to stabilize, with minimal further leakage.

In conclusion, the increased thermal energy may accelerate the molecular motion of curcumin, facilitating its release. However, this release does not qualify as a true thermo-responsive release since material disintegration and extensive drug release were not

achieved at 45°C conditions. As a result, we have removed the term "thermal responsive". Also, we provided an explanation for the heat-enhanced drug release.

Revision made:

To further estimate the pH-responsive drug releasing capacity, ${}^R\text{M}_{(I+C)}@CaP_{(p)}$ NPs (contained 0.4mg IR780, 0.2mg Cur and 200 μ M Partzyme) were conducted with release tests by dialysis membranes. Since IR780 generates heat under laser irradiation, it will promote the diffusion of drug molecules from the material, so different drug release environments with pH and temperature were designed (pH=7, 37°C; pH=7, 45°C; pH=6, 37°C; pH=6, 45°C). Therapeutic agents were observed through UV-Visible spectroscopy, at 780nm, 425nm and 560nm respectively (**Figure S7**). We have observed that at 45°C, there is an increased release of the drug. Approximately 35% IR780 and 43% Cur were released at pH=6, 45 °C, whereas only 9% IR780 and 14% Cur were released at pH=6, 37 °C. A similar phenomenon was also found in the pH=7 group. More than 20% IR780 and 30% of curcumin were released at pH=7 45°C. However, less than 5% of IR780 and 10% of curcumin were released at pH=7 at 37°C.

Question: In addition, the authors should clarify further on how drug release in different temperature is evidence of drug loaded inside micelles.

We thank the reviewer for raising this important question, and indeed, our explanation does have some logical confusions. In terms of drug loading locations, curcumin can be loaded within the calcium phosphate shell or within the micelles. From experimental results, under acidic pH conditions, the calcium phosphate shell can be degraded, promoting the complete release of DNAzyme. However, the curcumin is not released. Therefore, this suggested that the drug is not loaded into the calcium phosphate shell but rather within the micelles. We appreciate the reviewer's suggestion regarding this statement, as it will enhance the rigor of the article.

Revision made:

Furthermore, we observed a pronounced pH-responsive releasing profile for Partzyme, since CaP possessed excellent pH-sensitive degradation performance, and more than 90% of the Partzyme was released from the system after 4h incubation at pH=6 (**Figure 2M**, Partzyme). However, the degradation of CaP did not mediate significant release of Cur and IR780 at pH 6 compared to Partzymes loaded in the CaP shell, reflecting that IR780 and Cur were mainly loaded inside the micelles. Subsequently, 1 μ L of released mediums from each group were further incubated with 1 μ L miRNA-21 (10 μ M) and 2 μ L HSP70 mRNA (10 μ M). We found that the release mediums in the pH=6 groups efficiently cleaved HSP70 (**Figure 2N**). Rapid release of Partzyme ensured silencing of HSP70 before PTT treatment and prevented the previously mentioned hysteretic effect.

2) Statement in the manuscript: Meantime, Zn²⁺-substitute CaP(p) showed a strong positive charge (10.01 mV) due to the uncoordinated calcium and zinc ions on its surface.

Question: Authors are requested to provide further clarification and data to support the presence of Zn on the biomaterials surface.

We appreciate the reviewer for raising this question regarding the distribution of zinc ions on the surface of the biomaterial. Indeed, it is not enough to determine the distribution of metal ions on the material's surface, solely based on the change in charge.

To address the reviewer's concern, we conducted further analyses, including elemental analysis (EDS) and elemental distribution (Mapping) analysis. We found that zinc ions and calcium ions are uniformly distributed throughout the entire material, suggesting the presence of zinc ions on the material's surface.

Furthermore, from a mechanistic perspective, the zeta potential is a parameter that characterizes the surface charge of nanomaterials, and because metal ions (zinc ions and calcium ions) in the CaP system carry positive charges, we infer that the positive charge is attributed to these two metal ions.

We hope that these experimental results and explanations address the reviewer's concerns.

Revision made:

Meantime, Zn²⁺-substitute CaP_(p) showed a strong positive charge (10.01 mV). Through the elemental mapping experiment, we found the Zn ions and Calcium ions are distributed throughout the NPs, hence the positive charge may be due to the uncoordinated calcium and zinc ions on its surface (Figure S4).

3) In Table S5 reports stability studies for biomaterials (Nano systems) at refrigerated and accelerated-temperature conditions.

Question: The zeta potential values are relatively closed to zero mV and are indicative of unstable nanoparticles. The authors should probe further on other factors that influence stability of nanoparticles.

The reviewer's comment is crucial, focusing on nanoparticle (NP) stabilization, which we've given thorough consideration. While a high zeta potential value generally indicates NP stability, there are additional factors to consider. These include PEG-mediated steric stabilization, Zn^{2+} -induced CaP stabilization, and their combined effect.

Steric stabilization, achieved through PEG, plays a significant role in enhancing NP stability. In our case, PEG forms a protective surface layer, effectively preventing NP aggregation. This steric stabilization arises from the interplay of electrostatic and steric repulsion forces between PEG chains, forming a barrier that prevents NPs from coming into close proximity. Even though the zeta potential value is slightly negative at -2.53 mV, it still contributes to electrostatic repulsion and overall stability.

Another aspect is Zn^{2+} -induced stabilization, where smaller zinc ions readily substitute calcium in amorphous CaP, inhibiting crystallization. This disruption of atomic order and decreased CaP solubility adds to the stability of the ${}^R M_{(I+C)} @ CaP_{(p)}$ nanoformulation at the particle level. Moreover, our experiments consistently demonstrated that the ${}^R M_{(I+C)} @ CaP_{(p)}$ nanoformulation retained its size in aqueous solution for over three month, further bolstering its stability.

In summary, our belief is that a combination of factors, including steric and electrostatic repulsion, Zn^{2+} -induced stabilization, collectively ensure the stability of the ${}^R M_{(I+C)} @ CaP_{(p)}$ nanoformulation.

Mentioned in the maintext:

Notably, ${}^R M_{(I+C)} @ CaP_{(p)}$ was found to be very stable (with same size) in aqueous solution for more than two weeks (Figure 2G), suggesting that this nanosystem possessed excellent stability, which may through a combination of Zinc additive and steric repulsion forces between the PEG chains.

Based on the aforementioned results, we conducted stability assessments on the final formulation of our nanomaterial in accordance with the guidelines provided by the International Council for Harmonisation of Technical Requirements for Pharmaceuticals for Human Use (ICH) **Table S5**. The stability studies were conducted for a duration of three months, and the results demonstrated the stability of the nanomaterial formulation during this period.

In addition, authors should perform statistical analysis for all the reported data (Table S5). Compare the data obtained at different storage temperature.

We appreciate the reviewer's attention to our data analysis. We have provided detailed original data from our experiments in the source data. We also compared the data between 40°C and 4°C. Furthermore, we conducted statistical analyses using Prism9 for each of the three data sets for every month, and the results showed no significant differences between the data sets.

Modified in the supporting information:

Stability studies for the ${}^R\text{M}_{(l+c)}@CaP_{(p)}$ Nano systems at refrigerator and accelerated temperature conditions.

Storage temperature	4°C				40°C			
	Initial	1	2	3	Initial	1	2	3
Particle size (nm)	99.5 ± 10.6	100.4 ± 15.7	101.5 ± 9.7	105.3 ± 11.2	101.3 ± 9.2 (ns)	103.4 ± 7.9 (ns)	104.6 ± 8.6 (ns)	105.8 ± 6.6 (ns)
PDI	0.12 ± 0.07	0.21 ± 0.05	0.19 ± 0.11	0.20 ± 0.12	0.16 ± 0.07 (ns)	0.12 ± 0.08 (ns)	0.19 ± 0.03 (ns)	0.21 ± 0.04 (ns)
Zeta potential (mV)	-2.53 ± 0.5	-3.1 ± 1.5	-3.6 ± 1.5	-4.2 ± 2.1	-2.2 ± 0.12 (ns)	-1.1 ± 1.3 (ns)	-1.5 ± 2.3 (ns)	-0.4 ± 0.8 (ns)

In our study, we investigated the stability of the formulation at refrigerator temperature (4–8 °C) and under accelerated conditions at 40 °C, by following with the ICH guidelines, specifically ICH Q1A(R2) for long-term stability testing. The results of these stability studies, as summarized in Table S5, indicate that there was no significant increase ($P > 0.05$) in particle size or polydispersity index (PDI) throughout the three-month experimental period. Moreover, we observed no significant deviations in the zeta potential, indicating the formulation's stability for a minimum of three months. **All the data were conducted statistical analyses using Prism 9, t-test.**

Raw data added in the source data

Size(nm)	month 0			month 1			month 2			month 3		
40°C	110.5	100.8	92.6	95.5	102.9	111.8	96	105.2	112.6	99.2	105.3	112.9
4°C	88.9	99.1	110.5	84.7	100.1	116.4	91.8	101.1	111.6	94.1	105.9	115.9
PDI												
40°C	0.09	0.17	0.22	0.2	0.11	0.05	0.21	0.19	0.23	0.15	0.21	0.27
4°C	0.05	0.11	0.2	0.15	0.22	0.26	0.09	0.19	0.2	0.09	0.2	0.31
Zeta mV												
40°C	-2.2	-2.2	-2.0	0.3	-1.1	-2.5	0.8	-1.6	-3.7	0.1	-0.5	-1.6
4°C	-2.01	-2.55	-3.03	-4.1	-3.1	-2.1	-2.9	-3.6	-4.3	-1.5	-4.2	-6.9

Reviewer #3

The authors have replied the comments that the reviewers addressed. It can be accepted for publication.

Reviewer #4 - New, PDAC therapy, preclinical models

This manuscript was responsive to the prior critiques from reviewers and seems to have addressed all comments. With regard to the in vivo models of pancreatic cancer used, the authors employ orthotopic implantation of human PDAC cell lines into the pancreas. These cells express luciferase, enabling effective in vivo imaging of tumors to verify their presence and impact of intervention. This approach has advantages given that the tumors are located in the natural organ, which is superior to subcutaneous models. The data presented are also transparent, showing each individual animal as a representative image and the error bars on graphs to capture variability. Some minor additions are recommended to improve the discussion/data on these in vivo models:

We sincerely appreciate the reviewer's comments of our manuscript. Your positive assessment and comments are highly encouraging to us. Your recognition of the in vivo models we used, involving the orthotopic implantation of human PDAC cell lines into the pancreas, is greatly appreciated. We agree that this approach, facilitated by luciferase-expressing cells for in vivo tumor imaging, offers distinct advantages by closely mimicking the natural organ environment, thus surpassing subcutaneous models. We are also grateful for your acknowledgment of the transparency in our data presentation, including individual animal representative images and the inclusion of error bars in graphs to capture variability.

1. Methods, more details related to the implantation of the cells into the pancreas is needed, or at least a reference to the paper from which they derived this method. This is usually done via a surgical procedure and this detail is not evident in the paper

We greatly appreciate your insightful comments, including your suggestion for more detailed information regarding the implantation of cells into the pancreas. You have raised a valid point, and we agree that providing additional information on the methodology for cell implantation is essential for the clarity and comprehensibility of our manuscript. In the experimental methods section, we provided a detailed description of the surgical procedure and appropriately referenced relevant literature to ensure the accuracy and traceability of our methods.

Revision made:

Orthotopic model construction procedure: The experimental procedure was based on previously reported methods³ with certain improvements. nude mice are anesthetized and positioned in a left lateral recumbent position. Following routine skin disinfection, a 1 cm oblique incision is made just below the right costal margin, allowing access to the abdominal cavity. Afterward, the abdominal cavity is opened, revealing the spleen and pancreas. A prepared cell suspension is then precisely injected into the pancreas, with attention to minimizing tissue damage. Hemostasis is addressed if necessary, and the incision is sutured

to close the abdominal cavity. Finally, the surgical site is disinfected once more, and the mice are placed in a warm incubator or recovery area for post-operative care, all while adhering to ethical and animal care guidelines.

2. Discussion, the authors should acknowledge the limitation of using immunodeficient mice to implant human tumors. Although necessary it does not likely recapitulate the immune features and desmoplasia of human tumors that are really important in this disease.

We appreciate the reviewer's insightful comments regarding the use of immunodeficient mice for implanting human tumors in our study. We agree that there are limitations associated with this model. It is essential to acknowledge that these models may not fully replicate the immune microenvironment and desmoplastic features observed in human tumors.

The host immune response plays a pivotal role in the development and progression of pancreatic cancer. Multiple studies have shown that immune system activation can lead to immune cell infiltration into tumors, resulting in tumor cell killing. In this experiment, due to our use of human-derived cell lines, we inevitably opted for immunocompromised nude mice as our research subjects. This is indeed a limitation of our study. In subsequent research, we plan to validate our findings in immunocompetent mouse models and further design and optimize our nanoparticles.

Furthermore, pancreatic cancer, characterized by its highly fibrotic nature, is another main reason for the lack of effective drug interventions in clinical practice. Humoral immunity also plays a significant role in tumor fibrosis. Encouragingly, our nanoparticle-mediated inhibition of HSP70 and PTEN is likely to attenuating the degree of fibrosis in pancreatic cancer. This, in turn, enhances the accessibility of the nanoparticle drug to tumor cells, thus improving therapeutic efficacy. However, as our experiment primarily focuses on the tumor dual-gene targeting capability of DNAzymes, hence, the immunotherapy focused investigation will be, and defiantly be the next step of the studies for our designed DNAzymes.

As mentioned by the reviewer, in the discussion section, we have incorporated a thorough description of the considerations associated with using immunodeficient mice for implanting human tumors. Additionally, we have introduced relevant references to support this perspective.

Revision made:

Conclusion

In conclusion, we have successfully constructed an intelligent nanomachine for multi-module synergistic photothermal therapy.....is expected to promote the application of MNAzyme in clinical treatment of cancer.

However, for pancreatic cancer treatment, in addition to the regulation of relevant oncogenes, the immune response is also a crucial aspect that cannot be overlooked⁴. Furthermore, the high degree of fibrosis is another reason for the lack of effective drug

interventions in clinical cancer treatment⁵. Therefore, in future experiments, utilizing immunocompetent animal models and observing whether our designed MNzyme system can activate the immune response, attenuate the fibrosis in pancreatic cancer, and consequently enhance the accessibility of nanoparticle drugs to tumor cells will be highly meaningful.

1. Kastantin, M., Ananthanarayanan, B., Karmali, P., Ruoslahti, E. & Tirrell, M. Effect of the Lipid Chain Melting Transition on the Stability of DSPE-PEG(2000) Micelles. *Langmuir* **25**, 7279-7286 (2009).
2. Kenworthy, A.K., Simon, S.A. & McIntosh, T.J. Structure and phase behavior of lipid suspensions containing phospholipids with covalently attached poly(ethylene glycol). *Biophys J* **68**, 1903-1920 (1995).
3. Wang, J. et al. Orthotopic and Heterotopic Murine Models of Pancreatic Cancer Exhibit Different Immunological Microenvironments and Different Responses to Immunotherapy. *Frontiers in Immunology* **13** (2022).
4. Cappellesso, F. et al. Targeting the bicarbonate transporter SLC4A4 overcomes immunosuppression and immunotherapy resistance in pancreatic cancer. *Nature Cancer* **3**, 1464-1483 (2022).
5. Zheng, D. et al. Biomimetic nanoparticles drive the mechanism understanding of shear-wave elasticity stiffness in triple negative breast cancers to predict clinical treatment. *Bioactive Materials* **22**, 567-587 (2023).